# Molecular basis of potent antiviral HLA-C-restricted CD8+ T cell response to an immunodominant SARS-CoV-2 nucleocapsid epitope

Yoshihiko Goto[1,2,16], You Min Ahn [3,4,16], Mako Toyoda[1], Hiroshi Hamana[5], Yan Jin[1], Yoshiki Aritsu[1], Takeshi Nakama[1], Yuka Tajima[1,2], Janesha C. Maddumage[3,4], Huanyu Li[1], Mizuki Kitamatsu[6], Hiroyuki Kishi [5], Akiko Yonekawa[7], Dhilshan Jayasinghe[3,4], Nobuyuki Shimono[7], Yoji Nagasaki[8], Rumi Minami[9], Takashi Toya[10], Noritaka Sekiya [11,12], Yusuke Tomita[2], Demetra S. M. Chatzileontiadou[3,4,13], Hirotomo Nakata[14], So Nakagawa [15], Takuro Sakagami[2], Takamasa Ueno [1], Stephanie Gras [3,4,13] ✉ & Chihiro Motozono [1] ✉

The emergence of SARS-CoV-2 Variants of Concern (VOC) is a major clinical threat; however, VOC remain susceptible to cytotoxic T lymphocyte (CTL) recognition. Therefore, it is crucial to identify potent CTL responses targeting conserved epitopes across VOCs. Here, we demonstrate that the nucleocapsid (N) protein induces efficient CTL responses in early pandemic COVID-19 convalescent donors. In the context of the *HLA-A24-B52-C12* haplotype, prevalent in Japan, the KF9 peptide (N$_{266-274}$: KAYNVTQAF) is immunodominant and restricted by HLA-C*12:02. KF9-specific T cells are cytotoxic and suppress viral replication of both the ancestral and multiple VOC SARS-CoV-2. KF9-specific CD8+ T cells maintain effector memory and terminally differentiated phenotypes for 12 months post-infection and proliferate rapidly upon recall. We also determine the structure of a TCR in the context of the HLA-C*12:02-KF9 complex, providing a prototype for the interaction of HLA-C with viral peptides. Surprisingly, despite the TCR's high affinity, the CDR3β loop almost lacks contact with the KF9 peptide. These findings highlight the importance of conserved epitopes and the role of HLA-C molecules in controlling SARS-CoV-2 VOC.

The T-cell immune response plays a vital role in the control of various viral infections. Initial studies suggested that SARS-CoV-2-specific T cell responses are correlated with COVID-19 severity[1,2]. This idea was subsequently explored in longitudinal studies in mild/severe or asymptomatic patients[3,4], and a phase I/II clinical trial is currently underway to assess the effectiveness and safety of HLA-matched SARS-CoV-2-specific T cells from recovered patients for treating severe COVID-19 patients[5]. Moreover, a recent study demonstrated that HLA-B*15:01 is associated with asymptomatic SARS-CoV-2 infection[6], indicating an important role of HLA-B*15:01-restricted CTLs in combating virally infected cells. So far, research has mainly focused on SARS-CoV-2-specific CTLs restricted by HLA-A or HLA-B molecules[7]. The role of HLA-C-restricted antigen-specific CTLs in COVID-19 and its association with disease progression is not well understood, although it is

reported that some HLA-C allotypes are associated with COVID-19 severity[8,9]. Due to the low expression levels of HLA-C compared to HLA-A and HLA-B molecules, about one-tenth[10,11], it was believed that HLA-C-restricted CTLs do not contribute to the control of viral replication. However, HLA-C*12:02-restricted HIV-specific CTLs have a potent ability to suppress HIV-1 replication[12], suggesting that HLA-C-restricted CTLs can play an important role in viral control.

In addition to Spike (S)-specific T cells, nucleocapsid (N)- and membrane (M)-specific T cell responses are associated with mild disease in convalescents[13,14] and HLA-B*07:02-restricted N-specific T cells control viral replication and are associated with less severity[15]. Given that the N protein is the most abundant viral protein in virally infected cell lines[16] and is highly conserved compared to the S protein[17], the strategy targeting N protein-derived antigens would be applicable for the development of a global vaccine against newly emerging SARS-CoV-2 VOCs[18].

In this study, we characterise CTL responses specific for S, N, and M antigens in early pandemic convalescents. We find that the N protein induces potent CD8+ T cell responses in vitro. We identify an immunodominant N-derived antigen, namely KF9/C12 restricted by HLA-C*12:02, and analyse the T cell responses in the context of the *HLA-A24-B52-C12* haplotype, which is present at a frequency of 8.4% in the Japanese population. Furthermore, we analyse the antiviral activity of HLA-C*12:02-restricted KF9-specific T cells against virally infected cells and its crystal structure elucidating the CD8+ T cell recognition of an HLA-C molecule, with some surprising features such an almost lack of contact by the CDR3β loop with the peptide, which differs from HLA-A- and HLA-B-restricted TCR recognition.

## Results

### SARS-CoV-2 nucleocapsid protein induced efficient T cell responses in early pandemic convalescent donors

We compared the T cell immune response against SARS-CoV-2 S, N, and M proteins from peripheral blood mononuclear cells (PBMCs) derived from early pandemic (non-vaccinated) COVID-19 convalescent donors ($n = 31$, Table 1) with overlapping peptide pools. Proliferating T cells were evaluated for upregulation of the activation markers CD25 and CD137 (Fig. 1a and Supplementary Fig. 1a), as previously described[19,20]. The percentages of CD25+CD137+CD8+ T cells after stimulation with the N peptides were significantly higher than those with the S and M peptide pools (Fig. 1b, ***$p = 0.0002$ and **$p = 0.001$ by Wilcoxon matched-pairs signed rank test; versus S and M pools, respectively). No significant difference was observed in CD4+ T cells after stimulation with the S, N, and M peptide pools (Fig. 1b). Since the N protein remained the most abundant viral protein in infected cell lines[16], we stimulated PBMCs with Beta-propiolactone (BPL)-inactivated virus particles to confirm whether the N protein can induce profound T cell responses (Fig. 1c)[21]. The percentages of CD25+CD137+CD8+ T cells after stimulation with the N peptide pools were significantly higher than those stimulated with S and M peptide pools (Fig. 1d, **$p = 0.0014$ and $0.0017$ by Wilcoxon matched-pairs signed rank test; versus S and M pools, respectively). In addition, N-reactive CD4+ T cells were more reactive than the S- and M-reactive ones (Fig. 1d, *$p = 0.0474$ and $0.025$ by Wilcoxon matched-pairs signed rank test; versus S and M pools, respectively). This suggests that inactivated virus particles induce more efficient N-reactive T cells than S- or M-reactive T cells, especially CD8+ T cells, in the early pandemic COVID-19 convalescent donors. These data indicate that, in the samples tested, the N protein can induce more robust CD8+ T cells than the S and M proteins upon SARS-CoV-2 infection.

### N protein-induced immunodominant responses in the context of the *HLA-A24-B52-C12* haplotype

We characterised the N-specific CD8+ T cell responses in the context of the *HLA-A24-B52-C12* haplotype, the most prevalent in Japan at a

**Table 1 | Clinical samples used in this study, related to Fig. 1**

| Donor ID | Sex | Age | COVID-19 severity | Sampling day post PCR positivity or onset | Blood collection |
|---|---|---|---|---|---|
| KK-003 | Female | 19 | Mild | 2 | 2020/7/21 |
| KK-005 | Female | 24 | Mild | 8 | 2020/7/22 |
| KK-006 | Female | 38 | Mild | 8 | 2020/7/22 |
| KK-007 | Female | 38 | Mild | 20 | 2020/8/26 |
| KK-008 | Male | 63 | Mild | 9 | 2020/8/11 |
| KK-009 | Male | 36 | Mild | 22 | 2020/8/12 |
| KK-010 | Male | 60 | Mild | 17 | 2020/8/14 |
| KK-011 | Female | 67 | Moderate | 28 | 2020/8/19 |
| KK-012 | Male | 35 | Moderate | 16 | 2020/8/31 |
| KK-013 | Male | 72 | Moderate | 11 | 2020/8/31 |
| TK-001 | Female | 62 | Mild | 28 | 2020/9/14 |
| TK-002 | Male | 60 | Mild | 25 | 2020/10/19 |
| TK-003 | Female | 75 | Mild | 28 | 2020/10/26 |
| IK-001 | Male | 45 | Mild | 6 | 2020/11/18 |
| IK-002 | Male | 42 | Mild | 4 | 2020/11/18 |
| IK-003 | Male | 38 | Mild | 109 | 2020/11/18 |
| IK-004 | Male | 53 | Moderate | 236 | 2020/12/3 |
| IK-005 | Male | 38 | Mild | 128 | 2020/12/7 |
| IK-006 | Male | 75 | Moderate | 10 | 2020/12/15 |
| IK-007 | Male | 77 | Moderate | 20 | 2020/12/9 |
| IK-008 | Male | 33 | Mild | 130 | 2020/12/9 |
| IK-009 | Male | 52 | Mild | 118 | 2020/12/15 |
| IK-010 | Male | 54 | Moderate | 8 | 2021/1/20 |
| IK-011 | Male | 66 | Moderate | 15 | 2021/1/20 |
| IK-012 | Male | 72 | Moderate | 8 | 2021/1/20 |
| IK-013 | Male | 70 | Moderate | 22 | 2021/1/20 |
| IK-014 | Male | 27 | Mild | 41 | 2021/2/16 |
| IK-015 | Female | 27 | Mild | 2 | 2021/3/10 |
| IK-016 | Male | 27 | Moderate | 68 | 2021/3/26 |
| IK-017 | Male | 39 | Mild | 173 | 2021/4/5 |
| IK-018 | Male | 48 | Mild | 154 | 2021/5/6 |

frequency of 8.4%[22]. PBMCs derived from COVID-19-recovered individuals carrying the *HLA-A24-B52-C12* haplotype ($n = 12$, Table 2) were tested for interferon-gamma (IFN-γ) release upon overlapping N-derived peptides presentation as previously described[23]. Two overlapping peptides (OLP) were screened per well to make the assays feasible with the limited PBMC available from each donor. We found that the peptide pools 11-12, 25-26, 65-66, and 67-68 showed better T cell responses (>80% responders) in convalescents with this haplotype (Fig. 2a). In particular, the stronger response to the sequential peptide pools 65-66 and 67-68 indicates the possibility of shared epitopes restricted by HLA-A*24:02, -B*52:01 or -C*12:02 allomorphs.

### KF9 is an immunodominant epitope restricted by HLA-C*12:02

To investigate the HLA restriction of the N-derived epitopes, we stimulated PBMCs from a convalescent donor, GV73, with N overlapping peptide pools, and CD8+ T cells were evaluated for the production of IFN-γ with peptide-pulsed target cells, C1R cells, C1R-A2402, C1R-B5202, and C1R-C1202 as previously described[19,20]. The production of IFN-γ was only detected in the presence of C1R-C1202 (Fig. 2b and Supplementary Fig. 1b), indicating that dominant T cell epitopes derived from the SARS-CoV-2 N protein are restricted by the HLA-C*12:02 molecule. A bioinformatic prediction study of potential SARS-CoV-2 T cell epitopes presented by HLA molecules commonly present in the Japanese population identified six candidate HLA-C*12:02-

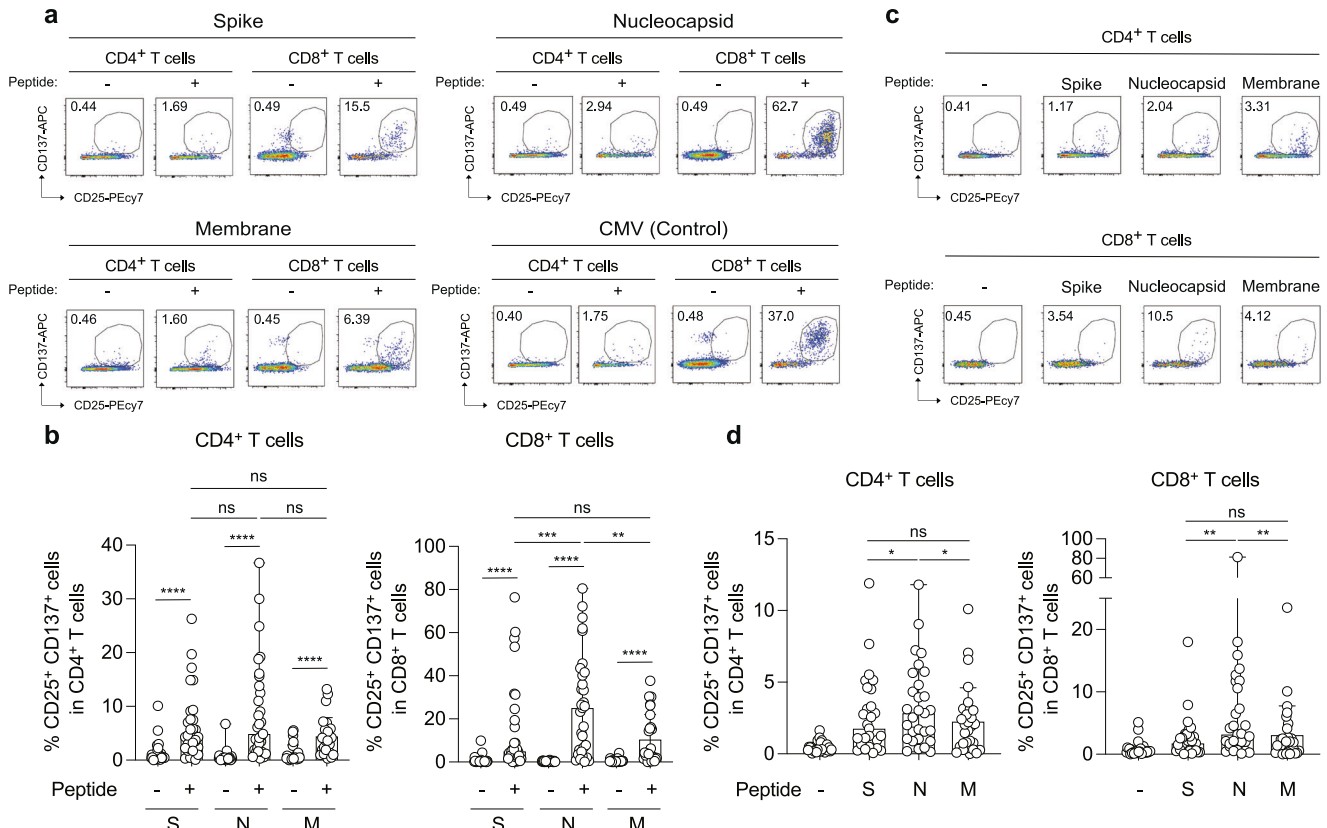

**Fig. 1 | Detection of SARS-CoV-2 S, N, and M-specific T cells stimulated with overlapping peptides and inactivated virus particles in convalescent donors. a**, **c** The numbers in the FACS plot represent the percentage of gated cells among CD4+ and CD8+ T cells. T cell activation toward CMV peptide pools is shown as a control. **b** The median percentages of CD25+CD137+CD4+ or CD8+ cells in convalescent donors (n = 31) and statistically significant differences in CD25+CD137+CD4+ or CD8+ cells (****p < 0.0001; versus no peptide) and in CD25+CD137+CD8+T cells after stimulation with the N pools between the indicated peptides (***p = 0.0002 and **p = 0.001; versus S and M pools, respectively) are determined by a two-tailed Wilcoxon matched-pairs signed-rank test. **d** The median

percentages of CD25+CD137+CD4+ or CD8+T cells in convalescent donors (n = 29), and statistically significant differences in CD25+CD137+CD4+T cells after stimulation with the N pools between the indicated peptides (*p = 0.0474 and *p = 0.025; versus S and M pools, respectively) and in CD25+CD137+CD8+T cells after stimulation with the N pools between the indicated peptides (**p = 0.0014 and **p = 0.0017; versus S and M pools, respectively) are determined by a two-tailed Wilcoxon matched-pairs signed-rank test. ns, no statistical significance. **b**, **d** Data are expressed as median. See also Supplementary Fig. 1a (Gating strategy). See also Table 1 (donor information).

binding peptides in the N protein: NTASWFTAL (NL9; residues 48-56), KAYNVTQAF (KF9; residues 266-274), FAPSASAFF (FF9; residues 307-315), SAFFGMSRI (SI9; residues 312-320), LTYTGAIKL (LL9; residues 331-339) and FSKQLQQSM (FM9; residues 403-411)[24]. Among the candidates, N-reactive T cells produced IFN-γ when C1R-C1202 were pulsed with the KF9 peptide (Fig. 2c). We confirmed that the KF9-stimulated T cells produced IFN-γ in the presence of C1R-C1202 but not of C1R-A2402 or C1R-B5202 (Fig. 2d). Of note, the KF9 peptide sequence is located in the immunogenic region of the nucleocapsid protein in convalescents carrying the *HLA-A24-B52-C12* haplotype (i.e., it is present in both the #66 (KRTAT<u>KAYNVTQAF</u>G) and #67 (T<u>KAYNVTQAF</u>GRRGP) 15-mer peptides and in peptide pools 65-66 and 67-68 (Fig. 2a). To assess the frequency of KF9/HLA-C*12:02-specific T cells ex vivo, we stained PBMCs with KF9/HLA-C*12:02 (KF9/C12) tetramers (Fig. 2e and Supplementary Fig. 1c). We found that KF9/C12-specific CD8+ T cells were detected in all HLA-C*12:02-positive donors tested (*n* = 14, Table 2) (Fig. 2e, median: 0.28%, ***p = 0.0002 by Mann–Whitney test; versus HLA-C*12:02-negative convalescents). These data indicate that KF9 is an immunodominant epitope presented by HLA-C*12:02 in COVID-19 convalescent donors.

It is reported that the HLA-C-restricted T cell epitopes are presented by other HLA-C allomorphs[25]. To explore the possibility that KF9 peptide is presented by other HLA allomorphs, we recruited 62 convalescents and examined the activated T cells in Activation-

Induced Marker assay. CD25+CD137+CD8+ T cells were significantly more abundant after stimulation with the KF9 peptide (median 9.02%) than without peptide (median 0.46%), showing in vitro proliferation in convalescents carrying HLA-C*12:02 (Fig. 2f, ****p < 0.0001 by Wilcoxon matched-pairs signed rank test). Interestingly, CD25+CD137+CD8+ T cells were significantly increased after stimulation with the KF9 peptide (median 0.58%) compared to a no-peptide control (median 0.44%) in a small number of donors (>1% CD25+CD137+CD8+ T cells were considered to be responders) without HLA-C*12:02 (Fig. 2f, **p = 0.0014 by Wilcoxon matched-pairs signed rank test). However, the level of the T cell activation was significantly lower than that in HLA-C*12:02-positive donors (Fig. 2f, p*** = 0.0002 by Mann–Whitney test; versus HLA-C*12:02-positive donors, indicating that the KF9 predominantly induces potent activation of CD8+ T cells in convalescents with HLA-C*12:02 in Japanese populations. To further investigate the possibility that the KF9 is presented by other HLAs, we focused on HLA-A, -B and -C alleles that the HLA-C*12:02-negative seven responders carry. An in silico analysis by NetMHCpan4.1 predicted that the KF9 showed strong binding potency to HLA-C alleles (Supplementary Table 1) (<0.5 rank% were considered as strong binders), suggesting that the KF9 is presented by other HLA-C alleles. Among them, we focused on HLA-C*14:02, at a frequency of 5.6% in Japan[24], most closely related to HLA-C*12:02, differing only five amino acids. In support of this, four out of seven donors (57.1%) were

**Table 2 | Clinical samples used in this study, related to Figs. 2, 5 and 6**

| Cohort | Donor ID | Sex | Age | No. of doses | Dose 1 | Dose 2 | Dose 3 | Days since last shot | COVID-19 severity | Sampling days post PCR positivity or onset | Blood collection | HLA-A | | HLA-B | | HLA-C | |
|---|---|---|---|---|---|---|---|---|---|---|---|---|---|---|---|---|---|
| C12 + COVID-19 convalescents | DC3 | Male | 61 | Not vaccinated | NA | NA | NA | NA | Moderate | 7 | 2022/3/9 | A*24:02 | A*26:03 | B*39:01 | B*52:01 | C*03:03 | C*12:02 |
| | DC6 | Female | 30 | x3 | BNT162b2 | BNT162b2 | BNT162b2 | NA | Mild | 10 | 2022/9/5 | A*24:02 | - | B*48:01 | B*52:01 | C*08:03 | C*12:02 |
| | IK-40 | Male | 31 | unknown | unknown | unknown | unknown | unknown | Mild | 2 | 2022/1/24 | A*24:02 | A*31:01 | B*35:01 | B*52:01 | C*03:03 | C*12:02 |
| | IK-63 | Female | 41 | x2 | unknown | unknown | NA | 195 | Mild | 17 | 2022/4/13 | A*24:02 | - | B*52:01 | - | C*12:02 | - |
| | GV33-1 | Male | 40 | x3 | BNT162b2 | BNT162b2 | mRNA-1273 | 477 | Mild | 9 | 2022/10/11 | A*24:02 | - | B*52:01 | B*59:01 | C*01:02 | C*12:02 |
| | GV37 | Female | 23 | Not vaccinated | NA | NA | NA | NA | Mild | 17 | 2021/5/20 | A*24:02 | - | B*52:01 | - | C*03:03 | C*12:02 |
| | GV66 | Male | 35 | x2 | BNT162b2 | BNT162b2 | NA | 176 | Mild | 15 | 2022/3/9 | A*02:01 | A*24:02 | B*40:06 | B*52:01 | C*08:01 | C*12:02 |
| | GV69 | Female | 46 | x3 | BNT162b2 | BNT162b2 | mRNA-1273 | 19 | Mild | 19 | 2022/4/18 | A*24:02 | - | B*15:01 | B*52:01 | C*04:01 | C*12:02 |
| | GV73 | Male | 42 | x3 | BNT162b2 | BNT162b2 | BNT162b2 | 145 | Mild | 14 | 2022/5/23 | A*11:01 | A*24:02 | B*52:01 | B*54:01 | C*01:02 | C*12:02 |
| | GV75 | Male | 38 | x3 | mRNA-1273 | mRNA-1273 | mRNA-1273 | 177 | Mild | 38 | 2022/12/5 | A*24:02 | - | B*52:01 | B*54:01 | C*01:02 | C*12:02 |
| | GV88 | Male | 43 | x3 | mRNA-1273 | mRNA-1273 | mRNA-1273 | 146 | Mild | 22 | 2022/9/3 | A*24:02 | - | B*52:01 | - | C*12:02 | - |
| | GV90 | Male | 64 | x3 | BNT162b2 | BNT162b2 | BNT162b2 | 153 | Mild | 16 | 2022/8/24 | A*24:02 | A*26:01 | B*40:02 | B*52:01 | C*12:02 | C*15:02 |
| | GV92 | Femlae | 33 | x3 | BNT162b2 | BNT162b2 | BNT162b2 | 130 | Mild | 22 | 2022/8/24 | A*24:02 | A*26:01 | B*15:01 | B*52:01 | C*03:03 | C*12:02 |
| | GV93 | Male | 33 | x3 | BNT162b2 | BNT162b2 | mRNA-1273 | 111 | Mild | 26 | 2022/8/27 | A*24:02 | - | B*07:02 | B*52:01 | C*07:02 | C*12:02 |
| | GV96 | Male | 41 | x3 | BNT162b2 | BNT162b2 | BNT162b2 | 135 | Mild | 26 | 2022/9/3 | A*02:01 | A*24:02 | B*52:01 | B*54:01 | C*01:02 | C*12:02 |
| | CAK-1 | Female | 86 | unknown | unknown | unknown | unknown | unknown | Moderate | 3 | 2022/1/31 | A*24:02 | - | B*13:01 | B*52:01 | C*03:04 | C*12:02 |
| | CAK-2 | Femlae | 84 | unknown | unknown | unknown | unknown | unknown | Mild | 3 | 2022/1/31 | A*24:02 | A*33:03 | B*44:03:01 | B*52:01 | C*12:02 | C*14:03 |
| | CAK-5 | Femlae | 68 | unknown | unknown | unknown | unknown | unknown | Mild | 3 | 2022/1/31 | A*24:02 | - | B*52:01 | - | C*12:02 | - |
| | CAK-6 | Male | 61 | unknown | unknown | unknown | unknown | unknown | Mild | 3 | 2022/1/31 | A*11:01 | A*24:02 | B*15:01 | B*52:01 | C*04:01 | C*12:02 |
| | CAK-8 | Male | 49 | unknown | unknown | unknown | unknown | unknown | Severe | 7 | 2022/1/31 | A*02:07 | A*24:02 | B*46:01 | B*52:01 | C*01:02 | C*12:02 |
| | CAK-18 | Female | 72 | x2 | unknown | unknown | NA | unknown | Moderate | 2 | 2022/2/16 | A*02:01 | A*24:02 | B*35:01] | B*52:01 | C*03:04 | C*12:02 |
| | CAK-21 | Male | 74 | Not vaccinated | NA | NA | NA | NA | Severe | 9 | 2022/3/2 | A*24:02 | - | B*40:01 | - | C*03:04 | C*12:02 |
| | CAK-24 | Male | 34 | Not vaccinated | NA | NA | NA | NA | Severe | 18 | 2022/3/2 | A*02:01 | A*24:02 | B*52:01 | B*67:01:02 | C*07:02 | C*12:02 |
| | AK-20 | Female | 58 | Not vaccinated | NA | NA | NA | NA | Severe | 10 | 2021/9/3 | A*24:02 | - | B*52:01 | - | C*12:02 | - |
| C12- COVID-19 convalescents | GV36 | Male | 40 | x3 | BNT162b2 | BNT162b2 | mRNA-1273 | 160 | Mild | 15 | 2022/8/3 | A*02:06 | A*24:02 | B*07:02 | B*51:01 | C*07:02 | C*14:02 |
| | GV61 | Female | 39 | x2 | mRNA-1273 | mRNA-1273 | NA | 159 | Mild | 20 | 2022/3/2 | A*24:02 | - | B*07:02 | B*54:01 | C*01:02 | C*07:02 |
| | GV63 | Male | 34 | x3 | BNT162b2 | BNT162b2 | BNT162b2 | 61 | Mild | 16 | 2022/2/24 | A*02:01 | A*24:02 | B*54:01 | - | C*01:02 | - |
| | GV64 | Female | 67 | Not vaccinated | NA | NA | NA | NA | Moderate | 24 | 2022/3/2 | A*24:02 | - | B*40:02 | B*54:01 | C*01:02 | C*03:03 |
| | GV65 | Male | 41 | x3 | BNT162b2 | BNT162b2 | NA | 318 | Mild | 10 | 2022/3/3 | A*24:02 | A*26:01 | B*40:02 | B*51:01 | C*15:02 | - |
| | GV82 | Male | 45 | x3 | BNT162b2 | BNT162b2 | BNT162b2 | 215 | Mild | 24 | 2022/8/10 | A*11:01 | A*24:02 | B*15:18 | - | C*07:04 | C*08:01 |
| | GV83 | Male | 38 | x3 | BNT162b2 | BNT162b2 | BNT162b2 | 125 | Mild | 24 | 2022/8/12 | A*11:01 | A*26:01 | B*35:01 | B*40:02 | C*03:04 | - |
| | GV84 | Male | 30 | x3 | mRNA-1273 | mRNA-1273 | mRNA-1273 | 104 | Mild | 16 | 2022/8/12 | A*11:01 | A*33:03 | B*44:03 | B*67:01:01 | C*07:02 | C*14:03 |
| | GV85 | Female | 46 | x3 | BNT162b2 | BNT162b2 | BNT162b2 | 228 | Mild | 13 | 2022/8/12 | A*02:01 | A*26:01 | B*40:02 | B*54:01 | C*01:02 | C*03:04 |
| | GV86 | Male | 63 | x3 | mRNA-1273 | mRNA-1273 | mRNA-1273 | 125 | Mild | 32 | 2022/8/20 | A*11:01 | A*24:02 | B*44:03 | B*67:01:01 | C*07:02 | C*14:03 |
| | GV87 | Female | 51 | x3 | mRNA-1273 | mRNA-1273 | mRNA-1273 | 139 | Mild | 23 | 2022/8/20 | A*02:06 | A*26:03 | B*35:01 | B*51:01 | C*03:03 | C*03:04 |
| | GV89 | Female | 46 | x3 | mRNA-1273 | mRNA-1273 | mRNA-1273 | 202 | Mild | 29 | 2022/8/23 | A*02:01 | A*24:02 | B*51:01 | B*67:01:01 | C*07:02 | C*14:02 |
| | GV91 | Female | 62 | x3 | BNT162b2 | BNT162b2 | BNT162b2 | 138 | Mild | 15 | 2022/8/24 | A*02:06 | A*26:01 | B*15:01 | - | C*03:03 | C*04:01 |
| | GV94 | Female | 46 | x3 | BNT162b2 | BNT162b2 | BNT162b2 | 203 | Mild | 24 | 2022/9/3 | A*24:02 | A*31:01 | B*15:01 | B*51:01 | C*04:01 | C*14:02 |
| | GV95 | Female | 41 | x3 | BNT162b2 | BNT162b2 | BNT162b2 | 146 | Mild | 25 | 2022/9/3 | A*02:06 | - | B*15:01 | B*59:01 | C*01:02 | C*14:02 |
| | GV97 | Male | 39 | x3 | BNT162b2 | BNT162b2 | BNT162b2 | 251 | Mild | 15 | 2022/9/6 | A*02:01 | A*24:02 | B*15:01 | B*46:01 | C*01:02 | C*03:03 |
| | GV98 | Female | 41 | x3 | BNT162b2 | BNT162b2 | BNT162b2 | 226 | Mild | 16 | 2022/9/7 | A*11:01 | A*33:03 | B*44:03:01 | B*54:01 | C*01:02 | C*14:03 |

**Table 2 (continued) | Clinical samples used in this study, related to Figs. 2, 5 and 6**

| Cohort | Donor ID | Sex | Age | No. of doses | Dose 1 | Dose 2 | Dose 3 | Days since last shot | COVID-19 severity | Sampling days post PCR positivity or onset | Blood collection | HLA-A | HLA-B | HLA-C |
|---|---|---|---|---|---|---|---|---|---|---|---|---|---|---|
| | IK-33 | Male | 41 | unknown | unknown | unknown | unknown | unknown | Mild | 34 | 2021/9/6 | A*26:01 / A*31:01 | B*40:01 / B*40:02 | C*03:03 / C*07:02 |
| | IK-34 | Male | 38 | unknown | unknown | unknown | unknown | unknown | Mild | 46 | 2021/9/16 | A*02:06 / A*02:07 | B*07:02 / B*46:01 | C*01:02 / C*07:02 |
| | IK-36 | Male | 61 | unknown | unknown | unknown | unknown | unknown | Mild | 2 | 2022/1/28 | A*03:01 / A*24:02 | B*15:01 / B*44:02 | C*03:03 / C*05:01 |
| | IK-41 | Female | 29 | unknown | unknown | unknown | unknown | unknown | Mild | 1 | 2022/1/26 | A*24:02 / A*31:01 | B*35:01 / - | C*03:03 / - |
| | IK-46 | Male | 25 | x2 | unknown | unknown | NA | unknown | Mild | 26 | 2022/2/9 | A*24:02 / A*33:03 | B*54:01 / B*58:01 | C*01:02 / C*03:02 |
| | IK-47 | Male | 36 | x2 | unknown | unknown | NA | unknown | Mild | 10 | 2022/2/10 | A*02:01 / A*24:02 | B*15:01 / B*40:06 | C*08:01 / - |
| | IK-48 | Male | 75 | unknown | unknown | unknown | unknown | unknown | Mild | 12 | 2022/2/26 | A*24:02 / - | B*40:01/10 / B*52:01 | C*04:01 / C*12:02 |
| | IK-50 | Male | 63 | unknown | unknown | unknown | unknown | unknown | Moderate | 14 | 2022/2/14 | A*24:02 / A*26:03 | B*15:01 / B*46:01 | C*01:02 / C*03:03 |
| | IK-54 | Male | 35 | x2 | unknown | unknown | NA | unknown | Mild | 21 | 2022/2/25 | A*02:01 / A*24:02 | B*15:18 / B*54:01 | C*01:02 / - |
| | IK-56 | Male | 43 | x2 | unknown | unknown | NA | unknown | Mild | 29 | 2022/3/2 | A*02:01 / A*02:03 | B*46:01 / B*58:01 | C*01:02 / C*03:02 |
| | IK-57 | Male | 47 | Not vaccinated | NA | NA | NA | NA | Mild | 42 | 2022/3/10 | A*02:01 / A*33:03 | B*44:03 / B*46:01 | C*03:03 / C*14:03 |
| | CAK-3 | Male | 74 | unknown | unknown | unknown | unknown | unknown | Moderate | 4 | 2022/1/31 | A*02:06 / A*24:02 | B*51:01 / - | C*14:02 / - |
| | CAK-13 | Female | 88 | unknown | unknown | unknown | unknown | unknown | Mild | 9 | 2022/2/9 | A*11:01 / A*24:02 | B*54:01 / B*55:02 | C*01:02 / C*03:03 |
| | CAK-20 | Male | 79 | unknown | unknown | unknown | unknown | unknown | Moderate | 18 | 2022/3/2 | A*24:02 / A*33:03 | B*15:01 / B*55:02 | C*01:02 / C*04:01 |
| | DC-1 | Male | 44 | unknown | unknown | unknown | unknown | unknown | Mild | 7 | 2022/2/28 | A*24:02 / A*33:03 | B*07:02 / B*44:03:01 | C*07:02 / C*14:03 |
| | DC-2 | Male | 78 | unknown | unknown | unknown | unknown | unknown | Mild | 10 | 2022/3/3 | A*24:02 / - | B*07:02 / - | C*07:02 / - |
| | AK-18 | Female | 25 | Not vaccinated | NA | NA | NA | NA | Moderate | 10 | 2021/9/3 | A*24:02 / A*26:01 | B*40:02 / B*40:06 | C*03:04 / - |
| | AK-19 | Male | 47 | Not vaccinated | NA | NA | NA | NA | Severe | 11 | 2021/9/3 | A*11:01 / A*31:01 | B*35:01 / B*51:01 | C*03:03 / C*14:02 |
| | AK-24 | Female | 28 | Not vaccinated | NA | NA | NA | NA | Severe | 13 | 2021/9/8 | A*02:03 / A*24:02 | B*46:01 / B*48:01 | C*01:02 / C*08:01 |
| | AK-25 | Female | 42 | Not vaccinated | NA | NA | NA | NA | Severe | 7 | 2021/9/8 | A*24:02 / A*26:01 | B*40:06 / B*54:01 | C*01:02 / C*08:01 |
| | AK-32 | Male | 57 | Not vaccinated | NA | NA | NA | NA | Moderate | 10 | 2021/9/13 | A*02:01 / - | B*13:01 / B*15:01 | C*03:03 / C*03:04 |
| C12+ seronegative | GV13 | Male | 23 | x3 | BNT162b2 | BNT162b2 | mRNA-1273 | NA | NA | NA | 2022/3/11 | A*24:02 / A*26:02 | B*15:01 / B*52:01 | C*03:03 / C*12:02 |
| | GV19 | Male | 24 | x2 | BNT162b2 | BNT162b2 | NA | | | | 2021/6/30 | A*24:02 / A*26:01 | B*52:01 / B*59:01 | C*01:02 / C*12:02 |
| | GV20 | Female | 24 | x2 | BNT162b2 | BNT162b2 | NA | | | | 2021/6/30 | A*02:06 / A*24:02 | B*35:01 / B*52:01 | C*08:01 / C*12:02 |
| | GV22 | Male | 23 | x2 | BNT162b2 | BNT162b2 | NA | | | | 2021/6/30 | A*24:02 / - | B*51:01 / B*52:01 | C*12:02 / C*14:02 |
| | GV33-2 | Male | 40 | x3 | BNT162b2 | BNT162b2 | mRNA-1273 | NA | NA | NA | 2021/7/5 | A*24:02 / - | B*52:01 / B*59:01 | C*01:02 / C*12:02 |
| C12- seronegative | GV9 | Female | 24 | x2 | BNT162b2 | BNT162b2 | NA | NA | NA | NA | 2021/6/29 | A*24:02 / A*26:01 | B*40:01 / B*54:01 | C*01:02 / C*03:04 |
| | GV15 | Female | 23 | x2 | BNT162b2 | BNT162b2 | NA | | | | 2021/6/29 | A*24:02 / - | B*40:02 / B*54:01 | C*01:02 / C*15:02 |
| | GV17 | Male | 24 | x2 | BNT162b2 | BNT162b2 | NA | | | | 2021/6/29 | A*11:01 / A*26:01 | B*40:01 / B*54:01 | C*01:02 / C*04:01 |
| | GV72 | Female | 20 | x2 | mRNA-1273 | mRNA-1273 | NA | | | | 2022/5/10 | A*26:01 / A*33:03 | B*40:02 / B*44:03:01 | C*03:04 / C*14:03 |

NA, not applicable.

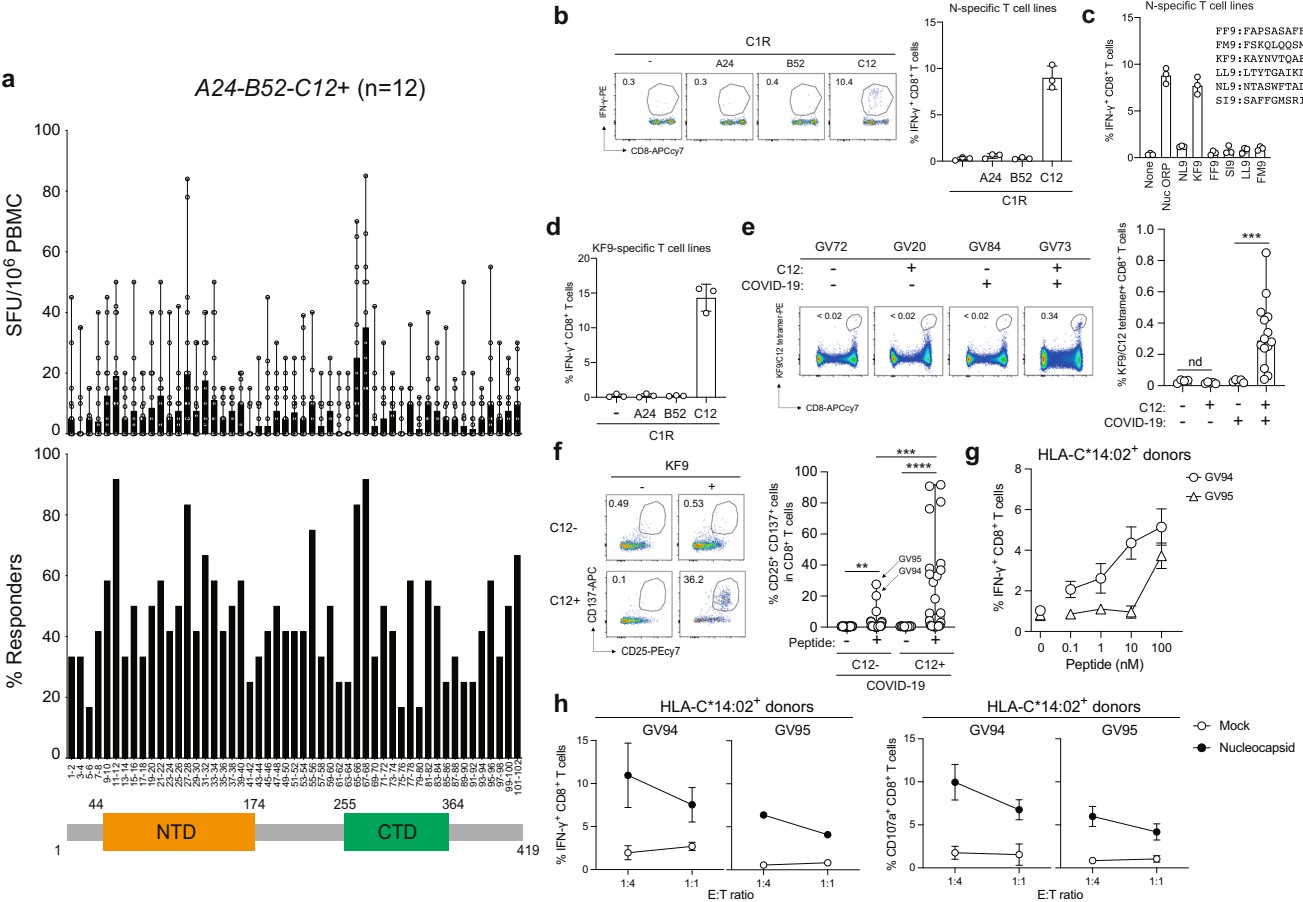

**Fig. 2 | SARS-CoV-2 N derived KF9 (N$_{266-274}$: KAYNVTQAF) is immunodominant in the context of HLA-C*12:02. a** The median frequency of magnitude of T cells response (upper graphs) and responders (lower graphs) to individual overlapping peptide (OLP) pairs in SARS-CoV-2 N proteins in twelve convalescents ($n = 12$) with a haplotype *HLA-A24-B52-C12*. NTD, N-terminal domain; CTD, C-terminal domain. **b** SARS-CoV-2 N-primed T cell lines of convalescents were stimulated with C1R-A2402, B5201, and C1202 pulsed with N overlapping peptides. Representative FACS plots showing intracellular expression of IFN-γ in the N-specific CD8$^+$ T cells in a donor (GV73). The percentage of IFN-γ$^+$ in N-reactive (**c**) or KF9-specific T cell lines (**d**) stimulated with C1R-C1202 pulsed with candidate peptides. **e** Detection of HLA-C*12:02-restricted KF9/C12-specific T cells in PBMCs by KF9/C12 tetramer. Representative FACS plots showing tetramer$^+$ CD8$^+$ T cells of an HLA-C12-negative donor (GV72), convalescent donor (GV84), and an HLA-C12-positive healthy donor (GV20) and convalescent donor (GV73). The median of the percentage of KF9/C12 tetramer$^+$CD8$^+$ T cells in HLA-C*12:02-positive twelve convalescent donors ($n = 12$) and a statistically significant difference is determined by unpaired Mann–Whitney

t-test (***$p = 0.0002$; versus HLA-C*12:02-negative convalescents. **f** Representative FACS plots showing surface expression of CD25 and CD137 on the CD8$^+$ T-cell subset of convalescents with or without HLA-C*12:02. The median of the percentage of CD25$^+$CD137$^+$ cells after stimulation with the KF9 peptide in HLA-C*12:02$^+$ convalescent donors ($n = 24$, median 9.02%) and HLA-C*12:02$^-$ convalescent donors ($n = 38$, median 0.58%). Statistically significant differences are determined by unpaired Mann–Whitney t-test in HLA-C*12:02-positive and negative convalescents (****$p < 0.0001$ and **$p = 0.0014$; versus no peptide, respectively) and CD25$^+$CD137$^+$CD8$^+$ cells in HLA-C*12:02-positive convalescents (***$p = 0.002$; versus HLA-C*12:02-negative convalescents). HLA-C*14:02-positive T-cell lines of convalescent donors (GV94 and GV95) were stimulated with A549-ACE/C1402 (Supplementary Fig. 1d) expressing nucleocapsid protein derived from prototype. The level of IFN-γ production of KF9-specific T cells in stimulation with the KF9 peptide (**g**) and N-expressing target cells (**h**), and the level of CD107a in response to those at different E:T ratios. **b–d**, **g**, **h** The assay was performed in triplicate, and the means are shown with the SD. **e**, **f** Data are expressed as median.

responders carrying HLA-C*14:02, and >20% of CD25$^+$CD137$^+$CD8$^+$ T cells were detected in two donors (GV94 and GV95) carrying HLA-C*14:02 (Fig. 2f and Table 2). Therefore, we hypothesized that the KF9 is also presented by HLA-C*14:02. To confirm this, we established T cell lines by stimulating PBMCs from two HLA-C*14:02$^+$ convalescents (GV94 and GV95) with the KF9 peptide. The resulting T cell lines were tested for their ability to recognise A549-ACE2-C1402 cells (Supplementary Fig. 1d) pulsed with the KF9 peptide or engineered to express the N protein from the prototype (D614G-bearing B.1.1 lineage). The KF9 peptide induced the production of IFN-γ by CD8$^+$ T cells in a dose-dependent manner (Fig. 2g). The level of IFN-γ and the degranulation surface marker CD107a among the CD8$^+$ T cell population treated with target cells expressing HLA-C*14:02 and N protein was higher than the mock and at all E:T ratios (Fig. 2h). These data suggest that the KF9 is presented, at least in part, by HLA-C*14:02 in Japanese convalescent donors.

## KF9/C12-specific T cells showed diverse TCR repertoire
We determined the KF9/C12-specific TCR repertoire ex vivo by single-cell sorting with the KF9/C12 tetramer in seven HLA-C*12:02$^+$ convalescent donors (Table 2 and Supplementary Fig. 1c) as previously described[20]. This analysis revealed that clonotypic diversity underpins the response to KF9/C12 epitope, with a total of 213 clonotypes isolated and 53 αβ TCR pairs (Supplementary Data 1). We obtained α-chain sequences from 3 donors and β-chain sequences from 7 donors. Three TRAV genes were shared in the three donors (GV37, GV66, DC3), namely TRAV5, TRAV6, and TRAV8. TRAV6 was used by the most frequent clonotype observed in donor DC3, and TRAV8 was used in the most frequently seen clonotype in donor GV66 (Supplementary Data 1). Similarly, the TRBV7 gene was observed in all samples ($n = 7$), while TRBV6 was observed in 85% of the samples (Supplementary Data 1). The length of CDR3α/β was highly variable, without bias or preference observed. Overall, the TCR repertoires were private, with

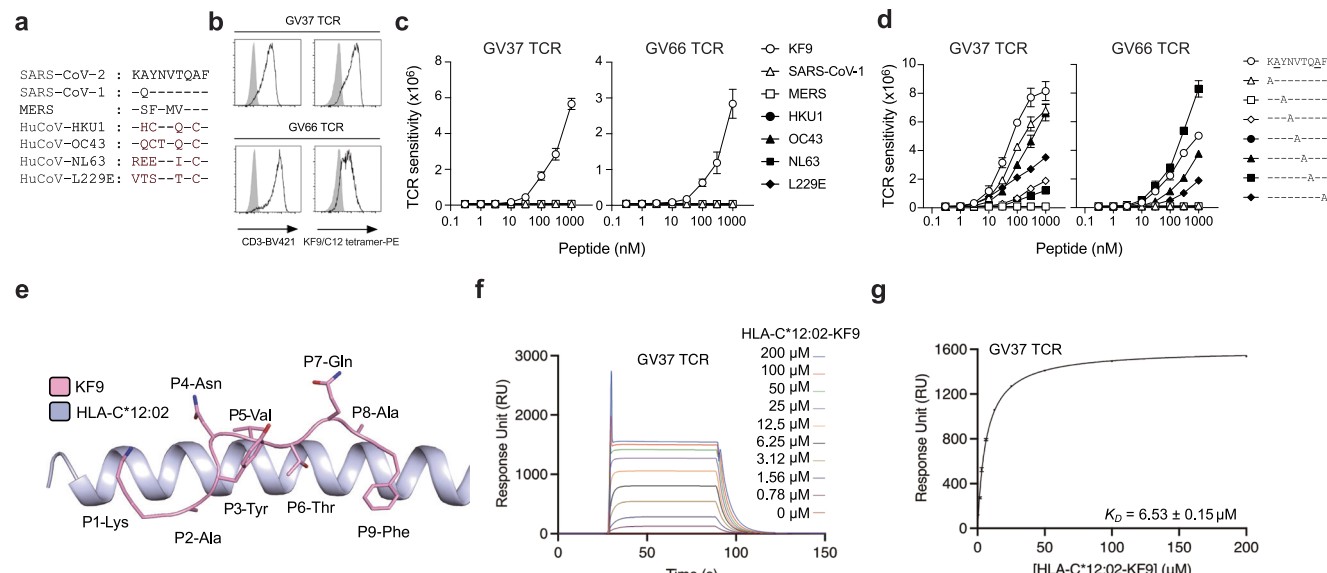

**Fig. 3 | KF9/C12-specific TCRs do not recognise homologous KF9 peptides derived from other coronaviruses and showed high affinity to the peptide from SARS-CoV-2. a** Sequence similarity of the KF9 peptide in SARS-CoV-1, MERS, and common cold coronaviruses. **b** Jurkat cells alone (shaded histogram) or those expressing KF9/C12-specific TCRs (GV37 and GV66) (open histogram) were stained with anti-CD3 mAb and KF9/C12 tetramer and then analysed by flow cytometry. **c** Peptide titration of TCR-transduced Jurkat cell lines. TCR cross-reactivity of TCR-reporter cells expressing KF9/C12-specific TCRs toward the variant peptides derived from SARS-CoV-1, MERS, and common cold coronaviruses. The assay was performed in duplicate, and the means are shown with the SD. Data are representative of three independent experiments. **d** The sensitivity of GV37 and GV66 TCRs toward a series of alanine substitutions in the KF9 peptide backbone except position 2 and position 8. The assay was performed in triplicate, and the means are shown with the SD. **e** Crystal structure of the HLA-C*12:02 (blue cartoon) presenting the KF9 peptide (pale pink sticks). **f** SPR sensorgrams for GV37 TCR with the analyte (HLA-C*12:02-KF9) flowed over immobilised GV37 TCR at concentrations ranging from 0 to 200 μM (curves of different colours). **g** Steady-state binding curve for GV37 TCR towards the HLA-C*12:02-KF9 complex. **e, g** Experiments were performed in duplicate with two independent biological samples, with the graph showing results from one experiment.

no public TCRs identified between individuals, suggesting that a diverse TCR repertoire underpins the immunodominant response observed in HLA-C*12:02⁺ convalescent donors.

## KF9/C12-specific TCRs do not recognise coronaviruses-derived homologous KF9 peptides

Previous studies show that a strong and dominant SARS-CoV-2-specific T cell response can be underpinned by the presence of pre-existing cross-reactive T cells from prior infections with coronaviruses[6,26]. SARS-CoV-1 and seasonal coronaviruses share a high protein sequence similarity with SARS-CoV-2, especially for the N protein (~90.5%). Therefore, we tested if KF9-specific T cells could recognise coronavirus-derived homologous peptides. Sequence alignment of the KF9 peptide showed that SARS-CoV-1-derived KF9 shared the highest sequence homology with only one residue difference at position 2 of the peptide (Fig. 3a). While the alignment showed that the KF9 homologous peptides all conserved Phe at position 9, one of the primary anchor residues[27], there was limited sequence conservation for the rest of the peptide with the seasonal coronavirus-derived peptides (Fig. 3a). Two KF9-specific TCRs expressing either the TRBV7 gene observed in all donors (GV37 TCR: TRAV5*01/TRBV7-8*01) or the TRBV6 paired with TRAV8 frequent in the sequenced clonotypes (GV66 TCR: TRAV8-6*02/TRBV6-5*01) were transduced into Jurkat cell lines (Fig. 3b). The Jurkat cell lines expressing either of the TCRs were stimulated with KF9 or the coronaviruses-derived homologues and showed that both TCRs were only able to recognise the SARS-CoV-2-derived KF9 peptide (Fig. 3c), and that even the closely similar SARS-CoV-1-derived KF9 peptide was not recognised by the TCRs. Moreover, we performed alanine scanning (excluding positions 2 and 8) using Jurkat cells expressing GV37 and GV66 TCR (Fig. 3d). GV37 TCR retained its reactivity toward alanine-substitutions at the N-terminus and position 6. In contrast, GV66 TCR showed comparable recognition of

position 7 but weak recognition of position 6. Both TCRs were not tolerant to substitutions in other central positions such as position 3, 4, and 5, indicating that P3-Tyr, P4-Asn and P5-Val are contacted by KF9-specific TCRs.

To better understand the selective T cell recognition of the SARS-CoV-2-derived KF9 peptide we refolded the peptide with HLA-C*12:02 and determined that the thermal stability of the complex (thermal melting temperature, $T_m$) was 57.63 ± 0.48 °C (Supplementary Fig. 1e). In addition, we also refolded HLA-C*14:02 with the KF9 peptide and show that the overall stability of the pHLA complex was 15 °C lower than the one of HLA-C*12:02-KF9 (Supplementary Fig. 1e), in line with the lower T cell activation observed as well (Fig. 2f). We solved the first crystal structure of the HLA-C*12:02 molecule in complex with the KF9 peptide at a high resolution of 1.61 Å (Table 3). The electron density was clear for the KF9 peptide in the antigen-binding cleft of HLA-C*12:02 (Supplementary Fig. 1f), which allowed an in-depth analysis of the peptide presentation. The structure shows that the cleft of HLA-C*12:02 is composed of a shallow B pocket, filled by tyrosine residues (Tyr7, Tyr9, Tyr67, and Tyr99) and a deep F pocket containing small hydrophobic residues (Leu95, Ser116, and Ile124) (Supplementary Fig. 1g). The molecular characteristics of the HLA-C*12:02 B and F pockets fit well with the peptide binding motif of the closely related allomorph HLA-C*12:03 (one residue difference, Arg97 to Trp97), showing a preference for P2-Ala and P9-Val/Tyr/Phe/Met/Ile[28]. The KF9 peptide docks well within HLA-C*12:02 with its anchor residues P2-Ala and P9-Phe (Fig. 3e), in addition to secondary anchor residue P6-Thr that further stabilises the peptide and interacts with the C/D pockets. As a result, P1-Lys, P4-Asn, P5-Val, P7-Gln, and P8-Ala are solvent exposed and could be contacted by TCRs (Fig. 3e). Among them, alanine scanning suggested that the exposed P4-Asn and P5-Val are potentially involved in the interaction with both GV37 and GV66 TCRs (Fig. 3d). In addition, as the B pocket is shallow and would favour the binding of small side-chain residues (Supplementary Fig. 1h), this could

**Table 3 | Data collection and Refinement statistics for HLA-C\*12:02-KF9 and GV37 TCR-HLA-C\*12:02-KF9**

| Data collection statistics | HLA-C\*12:02-KF9 | GV37 TCR-HLA-C\*12:02-KF9 |
|---|---|---|
| Space group | I 2 | P 2₁ |
| Cell Dimensions (a, b, c)(Å), angle (°) | 62.10 77.12 83.11 β = 100.20 | 75.01 59.47 109.85 β = 109.91 |
| Resolution (Å) | 47.90–1.61 (1.64–1.61) | 45.46 – 2.54 (2.66 – 2.54) |
| Total number of observations | 336491 (16729) | 202995 (22715) |
| Nb of unique observations | 49905 (2442) | 29890 (3299) |
| Multiplicity | 6.7 (6.9) | 6.8 (6.9) |
| Completeness (%) | 100.0 (100.0) | 98.6 (89.2) |
| $I/\sigma_I$ | 14.8 (3.7) | 9.6 (1.1) |
| $R_{pim}{}^a$ (%) | 3.2 (20.2) | 6.1 (52.3) |
| $CC_{1/2}$ | 0.997 (0.922) | 0.995 (0.691) |
| **Refinement Statistics** | | |
| $R_{factor}{}^b$ (%) | 20.4 | 21.1 |
| $R_{free}{}^b$ (%) | 23.1 | 25.8 |
| R.m.s.d. from ideality | | |
| Bond lengths (Å) | 0.004 | 0.002 |
| Bond angles (°) | 0.65 | 0.45 |
| Ramachandran plot (%) | | |
| Allowed | 99.73 | 92.98 |
| Disallowed | 0.26 | 0 |
| PDB Code | 9F13 | 9HLJ |

Values in parentheses are for the highest resolution shell. $CC_{1/2}$, correlation coefficient;

$^a$ $R_{pim} = \dfrac{\sum_{hkl} \sqrt{\frac{1}{n-1} \sum_{j=1}^n |I_{hkl,j} - \langle I_{hkl} \rangle|}}{\sum_{hkl} \sum_j I_{hkl,j}}$

$^b$ $R_{factor} = \dfrac{\sum_{hkl} |F_o - F_C|}{\sum_{hkl} |F_o|}$ for all data except » 5%, which were used for $R_{free}$ calculation.

suggest that the SARS-CoV-1-derived KF9 homologous peptide, with a larger P2-Gln residue, might not bind stably to HLA-C\*12:02. HLA stabilization assay revealed that SARS-CoV-1-derived KF9 homologous peptide (KF9-Q2) weakly bound to HLA-C\*12:02 (Supplementary Fig. 1i-k), and that it binds to HLA-C\*12:02 with a 10 °C lower Tm compared to KF9 derived from SARS-CoV-2 (Supplementary Fig. 1e), provides some basis for the lack of recognition by both the GV37 and GV66 TCRs (Fig. 3c).

### GV37 TCR recognises HLA-C\*12:02-KF9 with high affinity despite the almost lack of CDR3β loop interaction with the peptide

To determine how CD8[+] TCRs recognise the HLA-C\*12:02-KF9 complex, we expressed and refolded the GV37 and GV66 TCRs. Unfortunately, the GV66 TCR was not amenable to be properly refolded and was not pursued. We used the GV37 TCR to carry out surface plasmon resonance (SPR) (Fig. 3f and 3g). The GV37 TCR binds with a slow on-rate, characteristic of CD8[+] TCRs, and a slow off-rate, characteristic of high-affinity TCRs (Fig. 3f). Indeed, a $K_d$ of ~6 μM was calculated for the GV37 TCR binding to the HLA-C\*12:02-KF9 complex, which is in the upper range of CD8[+] TCRs[29].

We solved the crystal structure of GV37 TCR in complex with HLA-C\*12:02-KF9 at a resolution of 2.54 Å (Table 3), with clear density for the KF9 peptide (Supplementary Fig. 2a and 2b). Overall, the GV37 TCR adopted a canonical diagonal docking mode onto the HLA-C\*12:02-KF9 complex (Fig. 4a), with a total buried surface area (BSA) of 2,068 Å$^2$ that is in the range for TCR-pHLA complexes[29]. The TCRα chain contribution is slightly more (52%) than the β-chain (48%) to the TCR BSA, with the CDR3α having the largest contribution (24%), followed by the CDR3β (17.5%), CDR1α (16.5%), CDR2β (16%), framework β-chain (FWβ, 14%) from the flanking regions of the CDR2β, and

CDR2α (9%) (Fig. 4b and Table 4). The CDR1β did not contribute to the interaction.

All the CDR loops, besides CDR1β, interacted with the HLA-C\*12:02 molecule; CDR1α with the N-terminal part of the cleft, CDR3β with the kinked region of the α2-helix, and CDR2β/FWβ region with the C-terminal part of the cleft on the α1-helix (Fig. 4b and Table 4). The peptide was mainly contacted by three CDR loops (CDR1α, CDR3α, and CDR2β, Fig. 4b). The CDR1α loop interacted with P1-Lys of the peptide, a peptide residue rarely contacted by CD8[+] TCRs[29], and Asp26α and Ser28α completely shielded the side chain of P1-Lys by creating a network with both the peptide and the HLA (Fig. 4c). The CDR3α loop made most of the contacts with the peptide, with the loop stretching over the central part of the peptide and contacting from P4-Asn to P7-Gln (Fig. 4d and Table 4). At the C-terminal part of the peptide, the major interaction from the TCR β-chain is made by Gln50 of the CDR2β loop with P7-Gln (Fig. 4e and Table 4). This interaction was further stabilised by a hydrogen bond between Gln50β and Asn95α. The TCR Vβ chain binds to the HLA-C\*12:02-KF9 complex with a 40° angle, while the Vα is perpendicular to the HLA (Supplementary Fig. 2c). This docking angle favoured the interaction of the FWβ/CDR2β with the HLA α1-helix and seemed to push or "scoop" the CDR3β loop away from the peptide (Fig. 4e and Supplementary Fig. 2c). As a result, the CDR3β loop only makes a single hydrophobic contact with P7-Gln by Ala97β (cut-off of 4 Å is used in Table 4). This is striking, as the CDR3β loop interacts with the peptide in >96% of the 257 mouse and human TCRpMHC crystal structures solved (Supplementary Data 2)[29,30]. Only 10 TCRpMHC structures show no CDR3β-peptide contact and two more only show a single contact (cutoff <4 Å, Supplementary Data 2). From those complexes, three of them involved a reversed TCRs[31,32] and interestingly three other TCRs were specific to an HLA-C allomorph[33,34], like the GV37 TCR.

Despite the limited interaction of the CDR3β loop with the KF9 peptide, the GV37 TCR contacted all the surface-exposed KF9 residues and covered the peptide surface upon binding (Fig. 4b). Additionally, we compared the HLA-C\*12:02-KF9 structure before and after GV37 TCR binding, and strikingly, the peptide did not change conformation (root mean square deviation [r.m.s.d.] of 0.25 Å) (Supplementary Fig. 2d), while the HLA-C\*12:02 cleft was 1.1 Å narrower after TCR binding (Supplementary Fig. 2e). The lack of structural changes in the KF9 peptide is likely to underpin the fast-on rate and high affinity observed for the GV37 TCR (Fig. 3f).

The crystal structure of the GV37 TCR in complex with HLA-C\*12:02-KF9 gave the opportunity to further investigate the molecular basis underpinning the lack of GV37 TCR recognition for the KF9 homologous peptides (Fig. 3a–c). We first used TFold software[35] to predict the structures of each peptide in the cleft of HLA-C\*12:02 molecule (Supplementary Fig. 3). The KF9 derived from SARS-CoV-1 is only different from SARS-CoV-2-KF9 by one single residue at position 2, from a small Ala to a large Gln. TFold predicted the SARS-CoV-1-KF9 structure to be very similar to the crystal structure of SARS-CoV-2-KF9 (Supplementary Fig. 3). The large P2-Gln is unfavorable for the shallow B pocket of HLA-C\*12:02, and this is demonstrated by the drop of 10 °C in the pHLA complex thermal stability compared to P2-Ala SARS-CoV-2-KF9 peptide (Supplementary Fig. 1e). All the other KF9 homologues were predicted to adopt a different conformation from the one observed for the SARS-CoV-2-KF9 peptide (Supplementary Fig. 3), with some also containing an unfavorable large side chain residue at P2. The different peptide conformations, compared to SARS-CoV-2-KF9, are likely to be unfavorable to the GV37 TCR binding, as the TCR binds without SARS-CoV-2-KF9 conformational change, which might be a structural requirement for TCR recognition.

Overall, the lack of stability of the KF9 homologous peptides in the cleft of HLA-C\*12:02 and the different conformation adopted by the peptides likely underpin the lack of GV37 TCR recognition of the KF9 homologous peptides observed (Fig. 3a–c).

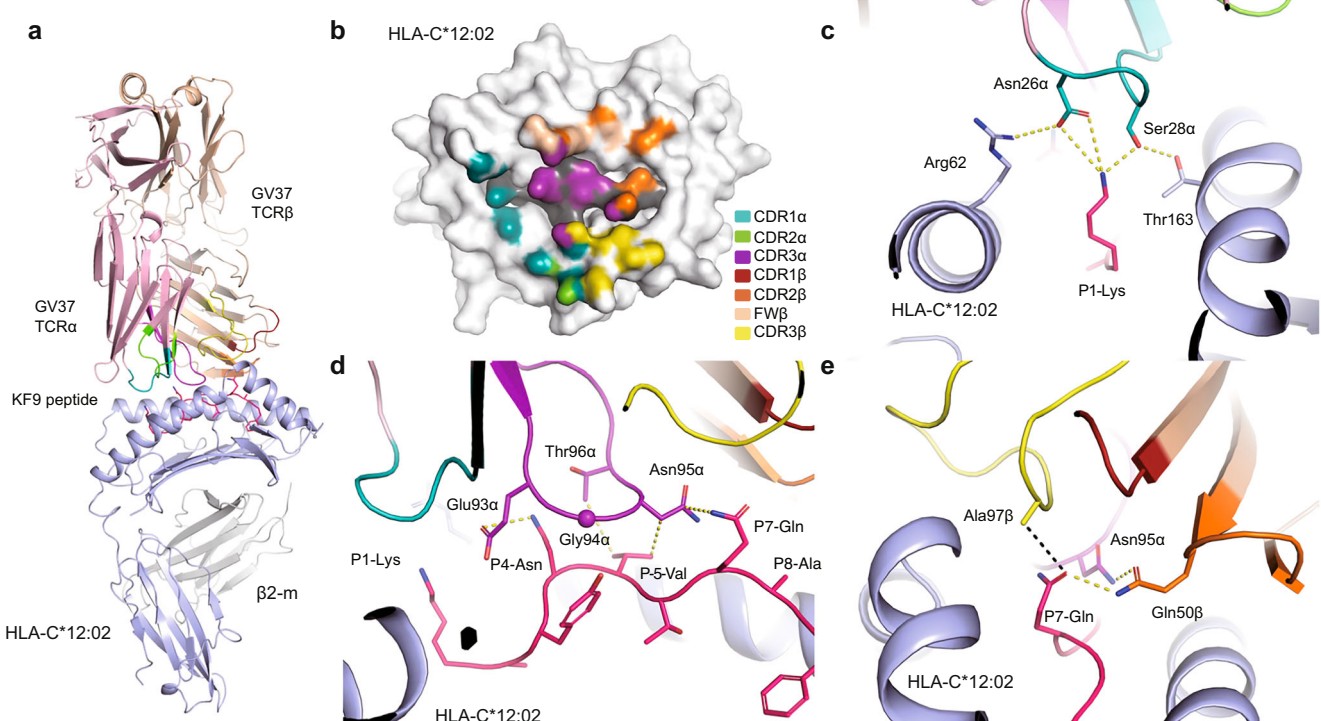

**Fig. 4 | Crystal structure of the GV37 TCR in complex with HLA-C*12:02-KF9.**
**a** Overview of the GV37 TCR-HLA-C*12:02-KF9 structure. HLA-C*12:02 is presented as a blue cartoon, the KF9 peptide as pink sticks, GV37 TCRα in pale pink, and the TCRβ in the beige cartoon. The CDR1α, CDR2α, and CDR3α loops are coloured in teal, chartreuse, and purple, respectively, while the CDR1β, CRD2β, CDR3β loops are coloured in red, orange, yellow, respectively. **b** GV37 TCR footprint on the surface of HLA-C*12:02 (white) and KF9 peptide (grey) colour-coded accordingly to the TCR segment involved in the interaction. **c-e** Cartoon representation of the GV37 TCR interacting with the KF9 peptide (pink stick), coloured as per panel (**a**), with yellow dashed lines representing contacts <4 Å, and **e** the black dashed line represents the unique contact of the CDR3β loop with the peptide.

## KF9-specific T cells efficiently suppress the replication of Omicron variants

To analyse the recognition of target cells expressing the N protein by the KF9/C12-specific T cells, we established T cell lines by stimulating PBMCs from three HLA-C*12:02+ convalescents (GV88, GV90, and GV96) with the KF9 peptide (Supplementary Fig. 4a and 4b). The resulting T cell lines were tested to recognise A549-ACE2-C1202 cells (Supplementary Fig. 4c) engineered to express the N protein from the prototype (D614G-bearing B.1.1 lineage). The KF9 peptide induced the production of IFN-γ by CD8+ T cells in a dose-dependent manner (Fig. 5a and Supplementary Fig. 4d). The EC$_{50}$ values of T cell line from GV88, GV90, and GV96 were $3.39 \pm 0.26$, $5.58 \pm 0.23$, and $6.53 \pm 0.97$ nM, respectively. To evaluate TCR avidity of GV37 TCR, we generated GV37 TCR-transduced CD8+ T cells by retroviral transduction and performed peptide titration (Supplementary Fig. 4e–g). The EC$_{50}$ value of the peptide was $4.41 \pm 0.3$ nM (Supplementary Fig. 4g), indicating that the sensitivity of GV37 TCR-transduced CD8+ T cells is comparable to that of parental T cell lines. The recognition of the prototype N protein by KF9/C12-specific T cells was observed at different E:T ratios (Fig. 5b and Supplementary Fig. 4h). Analysis of five additional donors confirmed these results (Fig. 5b, **p = 0.0078 by Wilcoxon matched-pairs signed rank test versus mock). Moreover, the cytotoxic potential of KF9-specific CD8+ T cells was also assessed by staining with CD107a. The percentage of CD107a+ CD8+ T cells (median 24.1%) among the CD8+ T cell population treated with target cells expressing the N protein was significantly higher than the mock (median 1.1%) (Fig. 5c and Supplementary Fig. 4i, **p = 0.0078 by Wilcoxon matched-pairs signed rank test versus mock). This suggests the cytotoxic potential of KF9-specific CD8+ T cells against target cells expressing the SARS-CoV-2 N protein.

We next investigated whether KF9/C12-specific T cells have a superior capacity to suppress the replication of prototype and Omicron variants such as BA.1, BA.2, and BA.5, by performing a T cell antiviral assay as previously described[20]. A549 cells expressing ACE2/C1202 were infected with SARS-CoV-2 viral variants and co-cultured with T cell lines specific for KF9/C12. T cell lines specific for QI9/A24 (S$_{1208-1216}$: QYIKWPWYI), an immunodominant and highly conserved epitope restricted by HLA-A*24:02, were included as a control in HLA-C*12:02+ donors (Supplementary Fig. 4j)[20,36]. It is noted that both of KF9/C12 and QI9/A24 epitopes are conserved in Wuhan, VOCs (Alpha, Beta, Gamma, and Delta), Omicron BA.1, BA.2, BA.5 and recent variants (Supplementary Fig. 5a). Moreover, we analysed amino acid mutation frequencies within the KF9 epitope using 5,378,593 SARS-CoV-2 genome sequences (see Materials and Methods for the details). The frequency of SARS-CoV-2 variants harbouring amino acid mutations within the KF9 epitope was less than 0.05% (Supplementary Fig. 5b), indicating that this region is highly conserved. Suppression of viral replication by T cells was evaluated by the amount of viral RNA in the supernatant at 72 h after infection. We confirmed that viral replication was comparable between the prototype and variants (Supplementary Fig. 5c). Both KF9- and QI9-specific T cells isolated from donors DC6 and GV69 suppressed replication of the prototype, Omicron BA.1, BA.2, and BA.5 comparably (Fig. 5d). KF9-specific T cells suppressed the replication more efficiently than the QI9-specific T cells at E:T ratio 1:1 (Fig. 5d). Therefore, we tested the antiviral activity of T cells from additional donors and confirmed that KF9-specific T cells (n = 6) could inhibit prototype and Omicron BA.1, BA.2 and BA.5 variants replication to a significantly greater extent than QI9-specific T cells (n = 5) (Fig. 5e and Supplementary Fig. 5d, **p = 0.0043 by Mann-Whitney test). These data indicate that HLA-C*12:02-restricted N-specific T cells have an

**Table 4 | GV37 TCR contacts with HLA-C*12:02 and the KF9 peptide**

| TCR segment | TCR residues | HLA-C*12:02 residue | Bond type |
|---|---|---|---|
| CDR1α | Thr25 | Arg62 | VdW |
| CDR1α | Asp26 | Arg62 | VdW |
| CDR1α | Ser28-Oγ | Thr163-Oγ1 | VdW, HB |
| CDR1α | Thr30 | Gln155, Ala158 | VdW |
| CDR1α | Tyr31-OH | Glu154-Oε1, Gln155 | VdW, HB |
| CDR2α | Phe50 | Glu154, Ala158 | VdW |
| CDR3α | Glu93 | Gln155 | VdW |
| CDR3α | Gly94-N-O | Gln155-Oε1 | VdW, HB |
| CDR3α | Thr96 | Arg69 | VdW |
| CDR3α | Pro97 | Arg69 | VdW |
| FWβ | Tyr48 | Arg69 | VdW |
| CDR2β | Glu52-Oε1 | Arg79-Nη1 | VdW, SB |
| CDR2β | Ala53 | Val76, Arg79 | VdW |
| CDR2β | Gln54 | Gln72 | VdW |
| FWβ | Leu55 | Arg69, Gln72, Ala73 | VdW |
| FWβ | Asp56-Oδ2 | Arg69-Nε-Nη2, Gln65-Nε2 | VdW, HB, SB |
| FWβ | Ser58-Oγ | Gln65-Nε2 | HB |
| CDR3β | Leu96 | Ala149 | VdW |
| CDR3β | Ala97 | Ala150 | VdW |
| CDR3β | Ala100 | Ala150, Gln155 | VdW |
| CDR3β | Gly101-O | Arg151-Nε | VdW, HB |
| CDR3β | Gly102 | Arg151 | VdW |
| CDR3β | Val103 | Ala149 | VdW |

| TCR segment | TCR residues | KF9 residue | Bond type |
|---|---|---|---|
| CDR1α | Asp26-Oδ1 | Lys1-Nζ | VdW, SB |
| CDR1α | Ser28-Oγ | Lys1-Nζ | VdW, HB |
| CDR3α | Glu93-O-Oε1 | Asn4-Nδ2 | VdW, HB |
| CDR3α | Gly94 | Val5 | VdW |
| CDR3α | Asn95-N-Nδ2 | Val5-O, Thr6-O, Gln7 | VdW, HB |
| CDR3α | Thr96 | Asn4 | VdW |
| CDR2β | Gln50-Ne2-Oε1 | Gln7-Oε1, Ala8 | VdW, HB |
| CDR3β | Ala97 | Gln7 | VdW* |

*CDR* Complementarity Determining Region, *FW* framework, *VdW* Van der Walls interaction, cut off 4 Å, *HB* Hydrogen bond, cut off 3.5 Å, *SB* salt bridge interaction cut off 5 Å. * VdW interaction at 4.01 Å (calculated by Contact software in CCP4).

enhanced capacity to recognise and suppress emerging SARS-CoV-2 Omicron variants compared to HLA-A*24:02-restricted S-specific T cells.

**Memory KF9/C12-specific T cells are retained for 6-12 months after infection and have the potency to proliferate upon recall in vitro**

We next investigated the durability of memory KF9/C12-specific T cells at 6-12 months post-infection. We used 7 additional donor samples to test the IFN-γ production at 6 months after infection using N overlapping peptides, showing that the peptide pool 65-66 was immunogenic in 6 out of 7 samples (Fig. 6a). In addition, KF9-reactive T cells were detected in all convalescent donors tested ($n = 7$) at 6 months after infection (Fig. 6a). This indicates that the KF9 epitope is still an immunodominant epitope in HLA-C*12:02+ convalescent donors even at 6 months after infection. Indeed, although the frequency of KF9/C12 tetramer+ T cell population was significantly reduced at 6 and 12 months, KF9/C12 tetramer+ T cells were still detectable (median 0.07% and 0.06%, respectively) in all convalescents tested ($n = 7$)

compared to HLA-C*12:02-negative convalescents (median 0.02 %) (Fig. 6b, *$p = 0.0167$ by Mann-Whitney test; versus HLA-C*12:02-negative convalescents). Moreover, the KF9/C12-specific T cells mainly retained a terminally differentiated (TEMRA) phenotype at 6-12 months (Fig. 6c, *$p = 0.0312$ by Wilcoxon matched-pairs signed rank test; versus TEM) while effector memory (TEM) phenotype was gradually decreased after 6-12 months (Supplementary Fig. 5e, *$p = 0.0156$ by Wilcoxon matched-pairs signed rank test; versus 1 month). These data indicate that KF9/C12-specific T cells retain a TEMRA phenotype 6-12 months post-infection.

To further assess the quality of long-lived memory KF9/C12-specific CD8+ T cells up to 12 months post-infection, we analysed their response upon recall. For that, we designed an in vitro memory T cell recall assay in which we stimulate CTV (Celltrace Violet)-labelled PBMCs with N overlapping peptides or the KF9 peptide, measuring the KF9-specific T cell proliferation capacity. The number of KF9-specific CD8+ T cells was significantly higher upon KF9 stimulation, after recall at 6- and 12- months post-infection, compared to that without stimulation (Fig. 6d and Supplementary Fig. 5f, *$p = 0.0156$ by Wilcoxon matched-pairs signed rank test; versus no peptide). In addition, the level of proliferated KF9-specific T cells upon N overlapping peptides was equivalent to that upon the KF9 stimulation, indicating that the KF9 is one of immunodominant epitopes among SARS-CoV-2 N protein in convalescents harbouring HLA-C*12:02 after 6- and 12-months. At one month after infection, there was a similar proportion of TEM and TEMRA T cells, while at 6 months post-infection, the majority of KF9-specific T cells showed TEMRA phenotype before stimulation (Fig. 6c). Moreover, while most KF9-specific T cells were TEMRA and TEM, the frequency of central memory (TCM) T cells was slightly increased at 12 months post-infection, and the TEM phenotypes were higher than TEMRA at 6 months post-infection (Fig. 6e, *$p = 0.0496$ by Wilcoxon matched-pairs signed rank test). These data confirmed that the KF9 epitope is immunodominant in convalescent HLA-C*12:02+ samples at 6–12 months after infection, but the phenotype of the proliferating KF9/C12-specific T cells was altered between 6- and 12-month post-infection. These data suggest that KF9/C12-specific CD8+ T cells are retained as long-term memory and have a potent proliferation capacity upon re-infection.

## Discussion

In this study, we identified an immunodominant epitope, the KF9/C12, derived from the N protein and restricted by HLA-C*12:02, and demonstrated that KF9-specific T cells display potent antiviral activity toward SARS-CoV-2 variants. The data suggest that, in addition to HLA-A and HLA-B-restricted T cells[15,37], HLA-C-restricted T cells may play an important role in controlling COVID-19. Although the expression level of HLA-C molecules is approximately one-tenth that of HLA-A or HLA-B molecules[10,11], HLA-C*12:02-restricted T cells have a potent ability to suppress HIV-1 replication and select HIV-pol escape mutation in HIV-1 infection[12]. Additionally, a whole-genome association study indicated that a variant located 35 kb upstream of the HLA-C gene (rs9264942) is associated with HLA-C mRNA expression and AIDS progression[38,39]. These studies suggest that, in addition to HLA-A and HLA-B, HLA-C-restricted immune responses play an important role in the control of viral infections. To our knowledge, this is the first report of the potent antiviral activity of HLA-C-restricted T cells in COVID-19. KF9/C12-specific T cells were maintained at a relatively high frequency of long-term memory at 6- and 12-month post-infection and showed efficient proliferation upon recall. Further studies are needed to confirm the association of HLA-C*12:02 with COVID-19 disease outcomes, but potentially, the memory CTLs could provide protection. This study also reports the first crystal structure of a CD8+ TCR engaging with a peptide-HLA-C*12:02 complex. The structure reveals some common features with previously solved TCR-pHLA structures in complex with HLA-A and HLA-B[29], but also highlights some striking differences. For

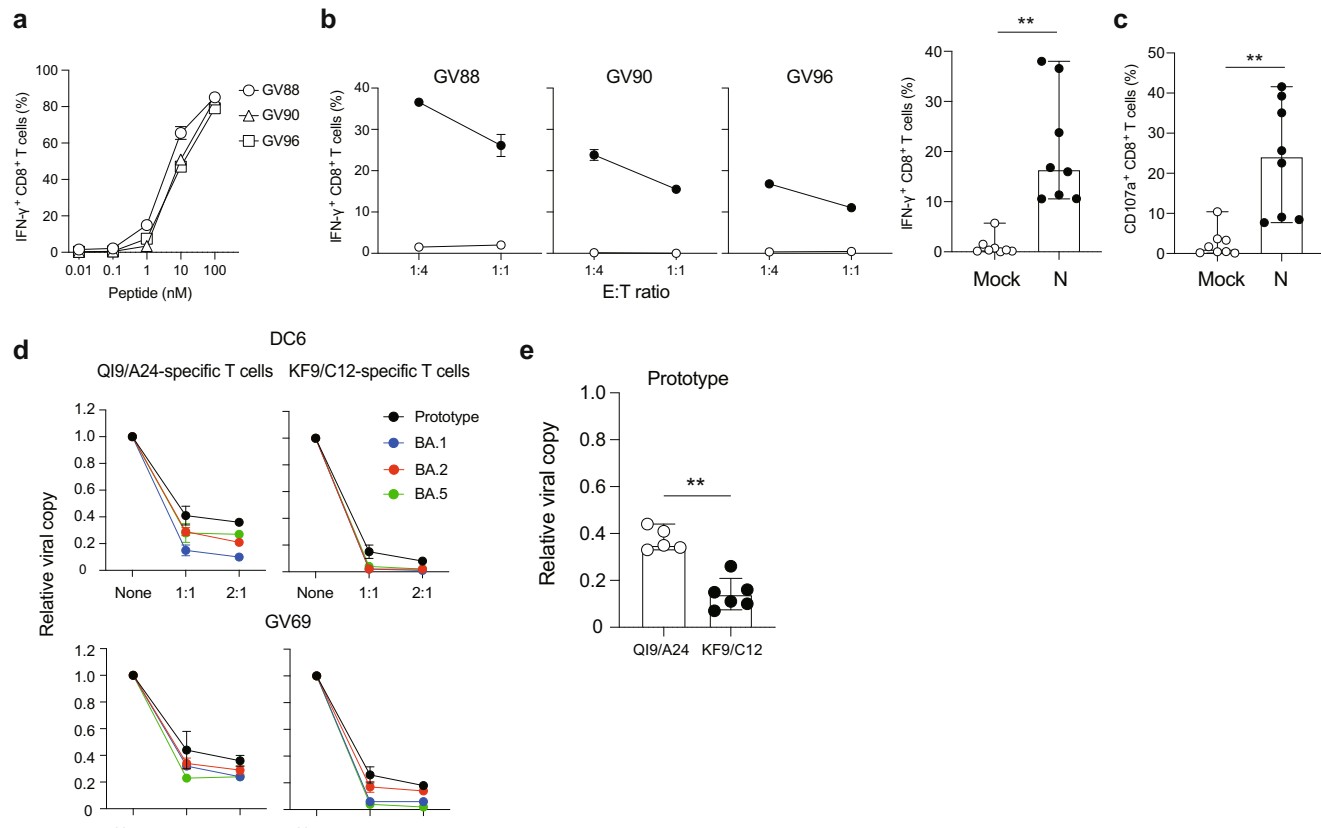

**Fig. 5 | T cell recognition to target cells expressing N protein and inhibition of SARS-CoV-2 viral replication by HLA-C12-restricted T-cell lines from convalescents. a–c** HLA-C12-positive T-cell lines of convalescent donors (Supplementary Fig. 4a and b) were stimulated with A549-ACE/C1202 (Supplementary Fig. 4c) expressing Nucleocapsid protein derived from prototype. The level of IFN-γ production of KF9-specific T cells in stimulation with the KF9 peptide (**a**, Supplementary Fig. 4d) and N-expressing target cells (**b**, Supplementary Fig. 4h) in three vaccinated donors and the level of CD107a in response to those (**c**, Supplementary Fig. 4i) in three convalescent donors. **b, c** A statistically significant difference is determined by Wilcoxon matched-pairs signed rank test (**p = 0.0078; versus mock). **d, e** Inhibition of viral replication of prototype, Omicron BA.1, BA.2 and BA.5 by KF9-specific T cells compared to that by SARS-CoV-2 S-derived QI9-specific T cells in two donors (**d**) and that of prototype by KF9-specific T cells in six donors and QI9-specific T cells in five donors (**e**) are shown. See Supplementary Fig. 4j (T cell lines tested in this assay), Supplementary Fig. 5c (Viral replication of the prototype and variants) and Supplementary Fig. 5d (Inhibition of viral replication of Omicron BA.1, BA.2 and BA.5 by KF9- and QI9-specific T cells). **e** A statistically significant difference is determined by unpaired Mann–Whitney t-test (**p = 0.0043; versus QI9-specific T cells). **a, b** The assay was performed in triplicate, and the means are shown with the SD. **c** The assay was performed in triplicate, and the median is shown. **d, e** Assay was performed in triplicate or quadruplicate, and the means are shown with the SD (**d**), and data are expressed as median (**e**).

example, while the buried surface area and contribution of the TCR segment were within the expected range[29], the almost complete lack of interaction of the CDR3β with the peptide was surprising and contrasts with previously reported structures of CD8+ TCRs. Interestingly, of the only five TCR-p-HLA-C crystal structures solved so far, three have no CDR3β-peptide contact, and one has no CDR3α-peptide contact. As there is a limited number of TCR-pHLA-C structures solved, it is unclear if this feature is common to HLA-C-restricted TCRs or specific to certain epitopes.

It has been reported that the HLA-B*07:02-restricted N-specific T-cell response is associated with less severe COVID-19 disease[15,37]. Moreover, in non-human primates, an important role of N-specific CD8+ T cells was observed in a study of an intranasal vaccine expressing viral non-S antigens[40]. These data indicate the importance of N-specific T cells in the control of COVID-19. In fact, the N protein remains the most abundant viral protein in virally infected cell lines[16]. In HIV-1 infection, the nucleocapsid protein of HIV-1 gag is expressed in infected cells at levels >1 log higher than other viral proteins[41,42]. Additionally, protective HLA-B*27:05 and HLA-B*57:01-restricted Gag-specific T cells were detected in HIV-infected individuals with long-term control[43–45]. Therefore, the most abundant and highly conserved SARS-CoV-2 N antigens are a promising target for a T-cell-induced vaccine against COVID-19. In support of this, we utilized BPL-

inactivated virus particles to confirm that the N protein can induce profound CD4+ and CD8+ T cell responses in vitro compared to the S and M proteins. Our data indicate that inactivated vaccines have the potency to induce better T cell responses specific for abundant immunogenic N antigens than non-N-derived antigens.

We previously demonstrated that SARS-CoV-2 S-derived antigens with L452R and L452Q mutations from Delta/Epsilon and Lambda variants, respectively, conferred escape from NF9 (S_{448-456}: NYNY-LYRLF)-specific T cell responses in the context of HLA-A*24:02, the most frequent HLA-I allomorph[19,46]. This has also been reported for other HLA-I molecules, such as HLA-A*02:01[47] and HLA-A*29:02[48]. Despite the accumulation of S mutations in the different SARS-CoV-2 variants, there is no complete immune escape from T cell recognition induced by infection or vaccination, due to the T cell responses toward conserved epitopes[20]. Moreover, the N protein is highly conserved among VOCs compared with the S protein[17]. Specifically, the KF9 epitope (N_{226-274}) is located in the C-terminal domain (CTD) region within the N protein, contributing to dimerization and RNA-binding[49,50]. This region is less prone to mutation as changes may impact viral replication. It would be interesting to determine whether a potent KF9/C12-specific T cell response against variants would be observed in convalescents infected with SARS-CoV-2 variants, which is associated with clinical outcome (severity, symptomatic or asymptomatic) in a future

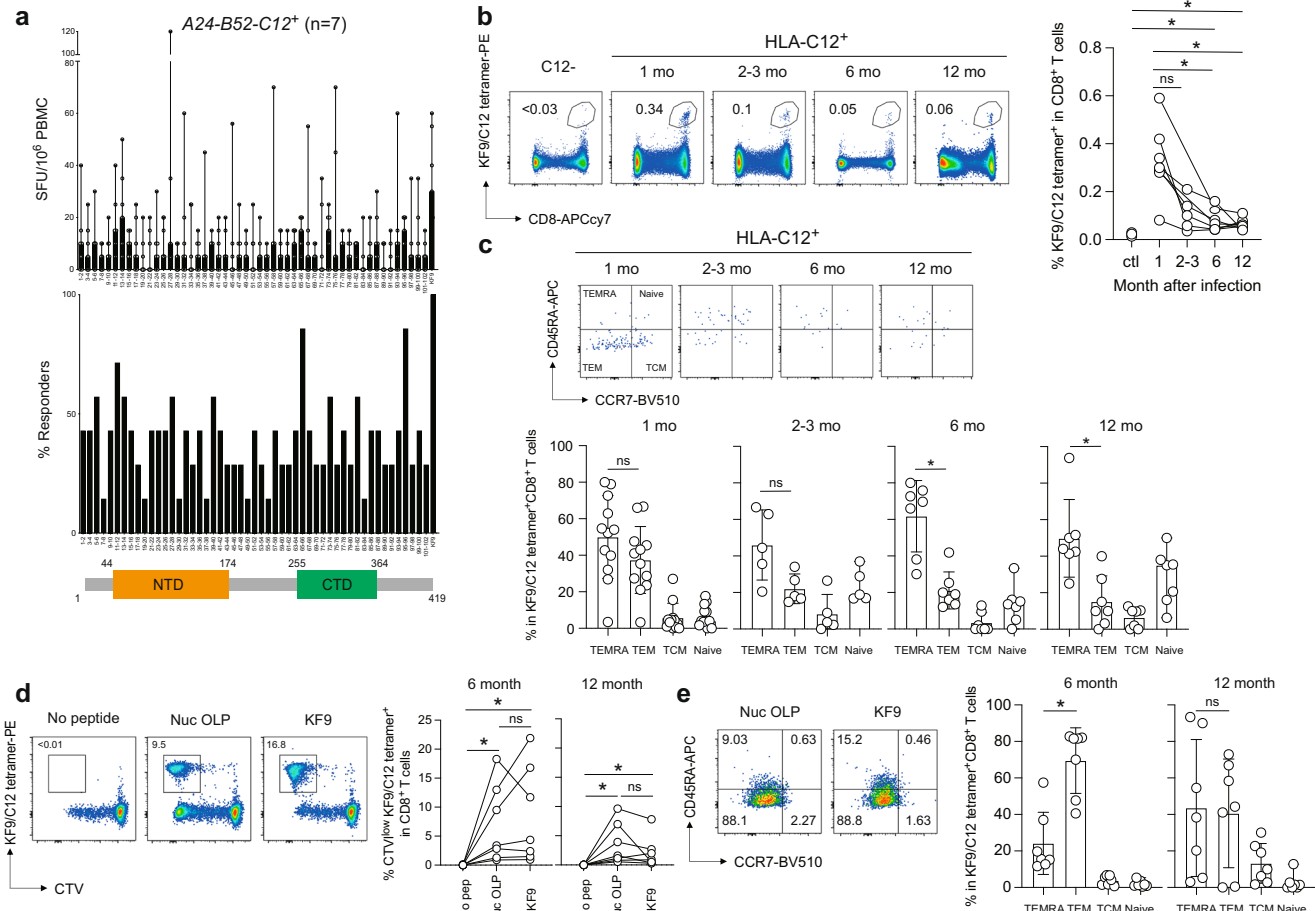

**Fig. 6 | KF9/C12-specific memory T cells are sustained for 6-12 months in COVID-19 convalescents. a** Frequency of magnitude of T cells response (upper graphs) and responders (lower graphs) to individual overlapping peptide (OLP) pairs in SARS-CoV-2 N proteins (2 μg/ml) or the KF9 peptide (100 nM) in seven COVID-19 convalescents (n = 7) with a haplotype *HLA-A24-B52-C12* 6 months after infection are shown. Data are expressed as median (upper panel). **b** Detection of HLA-C*12:02-restricted KF9/C12-specific T cells in PBMCs from seven COVID-19 convalescents (n = 7) by KF9/C12 tetramer. A statistically significant difference is determined by unpaired Mann–Whitney t-test (*p = 0.0167; versus HLA-C*12:02-negative convalescents). **c** Distribution of naïve, central memory (TCM), effector memory (TEM), and terminally differentiated effector memory cells (TEMRA)

among the KF9/C12-specific CD8+ T cells in PBMCs from seven COVID-19 convalescents (n = 7) and data are expressed as median. A statistically significant difference is determined by unpaired Mann–Whitney t-test (*p = 0.0312; versus TEM). See Supplementary Fig. 5f (Gating strategy). **d, e** Celltrace dye-labelled PBMCs were stimulated with N overlapping or KF9 peptides. The frequency (**d**) and differentiation status (**e**) of tetramer+ KF9/C12-specific CD8+ T cells were analysed by flow cytometry after 7 days of stimulation and data are expressed as median. Statistically significant differences are determined by Wilcoxon matched-pairs signed rank test in proliferating KF9/C12-specific T cells (*p = 0.0156; versus no peptide) (**d**) and the TEM phenotypes at 6 months (*p = 0.0496; versus TEMEA) (**e**).

study. Moreover, the strategy targeting conserved regions of viral proteins has been applied for T cell vaccines against viruses with high mutation rates, such as HIV-1[51]. Therefore, targeting more conserved viral proteins would be an applicable strategy for developing a global vaccine against newly emerging SARS-CoV-2 variants.

It has been reported that the HLA-B*07:02-restricted SPR/B7 (N$_{105-113}$: SPRWYFYYL)-specific T cell response is associated with less severe COVID-19[15], and the SPR/B7-specific T cells expressed a private TCR repertoire with a TRBV27 bias gene usage[52], consistent with previous studies[26,52]. In line with these findings, KF9/C12-specific T cells also showed diverse TCR clonotypes, suggesting a potential common feature of efficient N-specific T cell response in SARS-CoV-2 infection. However, their epitope sequence identities among seasonal coronaviruses differ between the SPR/B7 and KF9/C12 epitopes. While the SPR/B7 sequence is relatively conserved among seasonal coronaviruses (OC43 and HKU-1), with pre-existing cross-reactive memory T cells in unexposed HLA-B*07:02+ individuals[52], the KF9/C12 peptide differs from that of seasonal coronaviruses. In addition, the KF9/C12-specific TCRs did not recognise the homologous peptides derived from seasonal coronaviruses.

Memory CD8+ T cells are the source of long-term protective cellular immunity and are crucial in viral clearance upon re-infection. However, the mechanism underlying functional memory T cell formation and recall response associated with protective immunity in COVID-19 is not well defined in humans. Here, we found that KF9/C12-specific T cells were maintained for up to 12 months post-infection and showed TEMRA phenotypes. This correlates with previous findings that long-lived memory TEMRA T cells are induced and maintained by breakthrough infection[53–55]. However, proliferated KF9-specific memory T cells showed a TEM phenotype upon recall, indicating that TEM phenotypes of KF9-specific T cells at 6 months after infection have the potential for rapid proliferation in case of re-infection. The activation kinetics of S- and N-specific CD8+ T cells were broadly similar after breakthrough infection[37], suggesting that infection-induced N-specific T cells may have contributed to the control of breakthrough SARS-CoV-2 infection and, therefore, provide protection.

We showed that the KF9/C12-specific T cells activated after SARS-CoV-2 infection exhibit the characteristics of protective CD8+ T cells, such as proliferation, effector and memory phenotypes, and high affinity. This indicates that HLA-C-restricted N-specific T cells might

have an important role in clearing SARS-CoV-2 infection. Our work highlights that HLA-C molecules, in addition to HLA-A and HLA-B allomorphs, are contributing to the viral clearance, and also that targeting conserved epitopes between VOCs could provide an advantage for T cell-based vaccination that should include other proteins in addition to spike.

## Methods

### Ethics Statement

For the use of human specimens, all protocols involving human participants recruited at Kumamoto University, Kyushu Medical Centre, Kyushu University, Tokyo Metropolitan Cancer and Infectious Diseases Centre, Komagome Hospital were reviewed and approved by the Institutional Review Boards of Kumamoto University (approval numbers 2066 and 461). All human participants provided written informed consent.

### Collection of human PBMCs

Human PBMCs were obtained from thirty-one early pandemic convalescents (median age: 48, Range: 19–77, 74% male) (Table 1); twenty-four HLA-C*12:02$^+$ convalescents (median age: 42.5, Range: 23-86, 58.3% male), thirty-eight HLA-C*12:02$^-$ convalescents (median age: 43.5, Range: 25–88, 63.2% male), five HLA-C*12:02$^+$ seronegative (median age: 24, Range: 23–40, 80% male) and four HLA-C*12:02$^-$ seronegative donors (median age: 24, range: 20–24, 25% male) (Table 2). PBMCs were purified by a density gradient centrifugation using Ficoll-Paque Plus (GE Healthcare Life Sciences, Cat# 17-1440-03) and stored in liquid nitrogen until further use.

### Cell culture

A549 cells stably expressing human ACE2 and HLA-A*24:02-IRES-GFP or C*1202 or C*1402-IRES-NGFR were generated by electroporated transduction and were maintained in Ham's-F12 (Wako, Cat# 080-08565) containing 10% foetal bovine serum (FBS) (Sigma-Aldrich, Cat#172012-500 ML). C1R cells expressing HLA-A*24:02[56], -B*52:01[57], or -C*12:02[12] were maintained in RPMI 1640 medium (Thermo Fisher Scientific, Cat# 11875101) containing 10% FBS.

### Virus

Four clinically isolated SARS-CoV-2 lineages were used: SARS-CoV-2 Wuhan strain [SARS-CoV-2/Hu/DP/Kng/19-020 (DDBJ Accession ID: LC528232)] was provided by the Kanagawa Prefectural Institute of Public Health. B.1.1.529 (Omicron/BA.1) lineage [hCoV hCoV-19/Japan/TKYX00012/2021 (GISAID Accession ID: EPI_ISL_8559478)] and B.1.1.529 (Omicron/BA.5) lineage [hCoV-19/Japan/TKYS14631/2022 (GISAID Accession ID: EPI_ISL_12812500)] was provided by the Tokyo Metropolitan Institute of Public Health, Tokyo, Japan. B.1.1.529 (Omicron/BA.2) lineage [hCoV hCoV-19/Japan/TY40-385-P1/2022 (GISAID Accession ID: EPI_ISL_9595859)] was provided by the National Institute of Infectious Diseases, Tokyo, Japan.

### Analysis of SARS-CoV-2 genomic variation

To investigate genomic mutations in SARS-CoV-2, we downloaded 17,163,274 genome sequences from the GISAID database as of March 9, 2025[58]. Among these, 5,553,216 sequences containing no ambiguous or degenerate bases were obtained. After further filtering for sequences that 1) were derived from human hosts, 2) exceeded 29,000 nucleotides in length, and 3) exhibited less than 2% nucleotide divergence across the genome, a total of 5,378,593 genome sequences were retained for mutation analysis (accessible at https://doi.org/10.55876/gis8.250519sk). For each sequence, SGV-caller was used to identify genomic mutations at nucleotide and amino acid levels[59].

### ELISpot assay

Ex vivo IFN-γ ELISpot assay was performed using the following antibodies and reagents: anti-human IFN-γ mAb 1-D1K, purified (Mabtech, Code:3420-3-1000, 1/500 dilution), anti-human IFN-γ mAb 7-B6-1, biotinylated (Mabtech, Code: 3420-6-250, 1/2000 dilution), Streptavidin-ALP (Mabtech, Code:3310-8-1000), 25×AP Colour Development Buffer (BIORAD), AP Conjugated Substrate Kit (BIORAD, Cat# 1706432) according to the manufacturer's protocol. Pools of 15-mer peptides, overlapping by 11 amino acids and spanning the SARS-CoV-2 nucleocapsid protein (SCRUM Inc.). Briefly, after washing a Multi-Screen 96-well plate with sterilised PBS and blocked with RPMI 1640medium (Thermo Fisher Scientific, Cat# 11875101) containing 10% FBS, $2 \times 10^5$ defrosted PBMCs per well were pulsed with each SARS-CoV-2 prototype nucleocapsid overlapping peptide (2 µg/ml) and cultured for 20 hours. The responses were defined as the number of spot-forming cells (SFC)/$10^6$ calculated by ImmunoSpot (Cellular Technology Limited). When the number of spots for the negative control (no peptide) was 0, thresholds for positive responses were defined as more than 2 spots, and when spots were more than one spot for the negative control, the responses were defined as more than twice.

### Tetramer staining

PE-conjugated KF9 peptide-loaded HLA-C*12:02 tetramers were generated by NIH (National Institutes of Health). SARS-CoV-2-derived peptides-loaded MHC class I tetramers were generated by QuickSwitchTM Quant HLA-A*24:02 Tetramer Kit-PE (MBL International Corporation, Cat# TB-7302-K1) according to the manufacturer's protocol. PBMCs were stained with tetramers for 30 min at room temperature. After tetramer staining, surface staining with the following antibodies: CD3 BV421 (UCHT1, 1/50 dilution), CD8 APCcy7 (RPA-T8, 1/100 dilution), CD14 PerCP/Cy5.5 (HCD14, 1/100 dilution), CD19 PerCP/Cy5.5 (HIB19, 1/100 dilution), CCR7 BV510 (G043H7, 1/25 dilution), CD45RA APC (HI100, 1/100 dilution; Biolegend) was performed. Dead cells were stained with 7-aminoactinomycin D (Biolegend, Cat# 420404). After incubation for 20 min, the cells were fixed with 1% paraformaldehyde (Nacalai Tesque, Cat# 09154-85), and levels of protein expression were analysed by flow cytometry using a FACS Canto II (BD Biosciences) and Cytek Northern Lights (Cytek Japan) followed by analysis using FlowJo v10 software (BD Biosciences).

### Activation-Induced Marker Assay

An activation-induced marker assay was performed as previously described[19,20,46]. Briefly, human PBMCs were pulsed with 100 nM of the peptides (KF9: KAYNVTQAF, residues 266-274 of the SARS-CoV-2 nucleocapsid protein, and QI9: QYIKWPWYI, residues 1208-1216 of the SARS-CoV-2 spike protein; Genscript) maintained in RPMI 1640 medium (Thermo Fisher Scientific, Cat# 11875101) containing 10% FBS and 30 U/ml recombinant human IL-2 (Peprotec, Cat# 200-02) for 14 days. SARS-CoV-2 Wuhan strain ($5 \times 10^5$ PFU/ml) was inactivated with 0.5% Beta-propiolactone (Cat#H0168, Tokyo Chemical Industry Co., Ltd) at 56 °C 45 min as previously described[21]. Human PBMCs were stimulated with the peptide pools of SARS-CoV-2 spike (Miltenyi Biotec, Cat#130-126-701), nucleocapsid (Miltenyi Biotec, Cat#130-126-698,) and membrane protein (Miltenyi Biotec, Cat#130-126-703) or 3 µl of the inactivated virus particles and cultured for 14 days. The in vitro expanded CD4$^+$ or CD8$^+$ T cells (i.e., T-cell lines) were restimulated with or without the peptide. After incubation at 37 °C for 24 h, the cells were washed and surface stained with following antibodies: CD3 FITC (UCHT1, 1/100 dilution), CD8 APCcy7 (RPA-T8, 1/100 dilution), CD14 PerCP/Cy5.5 (HCD14, 1/100 dilution), CD19 PerCP/ Cy5.5 (HIB19, 1/100 dilution), CD25 PEcy7 (M-A251, 1/50 dilution) and CD137 APC (4B4-1, 1/50 dilution; Biolegend). Dead cells were stained with 7-aminoactinomycin D (Biolegend, Cat# 420404). After incubation for 20 min on ice, the cells were fixed with 1% paraformaldehyde (Nacalai Tesque, Cat# 09154-85), and levels of protein expression were analysed by flow cytometry using a FACS Canto II (BD Biosciences) and

Cytek Northern Lights (Cytek Japan) followed by analysis using FlowJo v10 software (BD Biosciences).

## Intracellular cytokine staining

Intracellular cytokine staining was performed as previously described[19,20,46]. Briefly, A549-ACE2-C1202 cells ($3 \times 10^5$ cells) were transfected with 2 μg of plasmids expressing prototype nucleocapsid using PEI Max (Polysciences, Cat# 24765-1) following the manufacturer's protocol. At 2 days post-transfection, the transfectants were harvested and mixed with the T-cell lines generated from HLA-C*12:02+ convalescents (see above) and cocultured with RPMI 1640 medium (Thermo Fisher Scientific, Cat# 11875101) containing 10% FBS, 5 μg/ml brefeldin A (Sigma-Aldrich, Cat# B7651) in a 96-well U plate at 37 °C for 5 h. The cells were washed and stained with the following antibodies: CD3 FITC (UCHT1, 1/100 dilution), CD8 APCcy7 (RPA-T8, 1/100 dilution), CD14 PerCP/Cy5.5 (HCD14, 1/100 dilution), CD19 PerCP/ Cy5.5 (HIB19, 1/100 dilution), and CD107a-BV421 (H4A3, 1/100 dilution; Biolegend). Dead cells were stained with 7-aminoactinomycin D (Biolegend, Cat# 420404). After incubation at 4 °C for 20 min, the cells were fixed and permeabilised with a Cytofix/ Cytoperm Fixation/Permeabilisation solution kit (BD Biosciences, Cat# 554714) and were stained with IFN-γ PE (4S.B3, 1/100 dilution; BD). After incubation at room temperature for 30 min, the cells were washed, and levels of protein expression were analysed by flow cytometry using a FACS Canto II (BD Biosciences) and Cytek Northern Lights (Cytek Japan), followed by analysis using FlowJo v10 software (BD Biosciences).

## TCR cDNA amplification from single T cells and construction of TCR expression vector

The cryopreserved PBMCs were stained with KF9/C12 tetramers PE (10 μg/ml), CD8 APCcy7 (RPA-T8, 1/100 dilution; Biolegend), and 7-amino-actinomycin D (7-AAD), and then tetramer+CD8+7-AAD- cells were sorted into 96-well plates (NIPPON Genetics, Cat# 4ti-0770/C) by using an FACS Aria II (BD Biosciences). TCRα and TCRβ cDNA pairs were amplified from single T cells by a one-step multiplex RT-PCR method described in our previous study[60]. The DNA sequences of the PCR products were then analysed by direct sequencing, and the TCR repertoire by IMGT/V-QUEST (https://www.imgt.org/IMGT_vquest/vquest). The amplified TCRα and TCRβ cDNA fragments were connected to the missing constant region and linked to the blasticidin S resistance (BlaR) gene by the Gibson assembly method with P2A ribosomal skipping sequences. The resultant TCRβ-P2A-TCRα-P2A-BlaR DNA was cloned into the PiggyBac vector (SBI, Cat# PB530A-2) by the Gibson assembly method.

## TCR sensitivity assay

The plasmid PB TCR-P2A-BlaR was electroporated into JurkatΔ-Luc with Transposase vector (SBI, Cat# PB200PA-1) using Neon® Transfection System (Thermo Fisher Scientific) under the condition 1200 v, 5 ms, 5 pulses. After 48 h, JurkatΔ-Luc cells stably expressing TCRs were selected with RPMI medium containing 10 μg/ml of blasticidin-S for 10-14 days. These cells were cocultured with A549-ACE2-C1202 cells expressing each spike protein with an E:T ratio of 2:1 and incubated with RPMI 1640 medium (Thermo Fisher Scientific, Cat# 11875101) containing 10% FBS at 37 °C for 6 h. The mixture was measured for luciferase production using a luminescent substrate (Promega, Cat#E2510) by a CentroXS3 plate reader (Berthold Technologies).

## The peptide-dependent stabilization assay

TAP-deficient MCF7 cells (MCF7ΔTAP1/2) were generated by using the CRISPR–Cas9 KO plasmid according to the manufacturer's instructions (Santa Cruz Biotechnology, Cat#sc-42981-SH and sc-42983-SH) at the University of Toyama. MCF7ΔTAP1/2 were enriched by staining the cells with an HLA-A, B, C mAb (1/50 dilution, W6/32, BioLegend, Cat#311410) and sorting HLA class I-negative cells using a FACSAria.

MCF7ΔTAP1/2 expressing HLA-C*12:02-IRES-NGFR MCF7ΔTAP1/2-C1202) were generated using Neon® Transfection System (Thermo Fisher Scientific) under the condition 1250 v, 20 ms, 2 pulses. A total of $1 \times 10^5$ cells of MCF7ΔTAP1/2-C1202 cells were pulsed with peptides and then incubated at 26 °C overnight as previously described[61]. At the end of the incubation, unbound peptides were removed, and cells were stained with Fixable Live/Dead Violet Dye (1/50 dilution, Invitrogen, Cat#L34964), PE-labelled HLA class I-specific mAb TP25.99 (1/50 dilution, Thermo Fisher Scientific, Cat#MA5-44116), APC-labelled NGFR mAb (1/50 dilution, BioLegend, Cat#345108) and analysed by Cytoflex (Beckman Coulter, Inc.), followed by analysis using CytExpert2.6 and FlowJo v10 software (BD Biosciences). The mean fluorescence intensity (MFI) was calculated, and the mean values of triplicates are presented. Relative HLA expression was calculated as the ratio of the MFI of peptide-pulsed MCF7ΔTAP1/2-C1202 cells to that of control (non-peptide-pulsed) cells kept at 26 °C.

## T-cell proliferation assay

T cell proliferation assay was performed using Cell Trace Violet Cell Proliferation Kit (Thermo Fisher Scientific, Invitrogen, C34557) following the manufacturer's protocol. Briefly, defrosted PBMCs were washed with sterilised PBS, and $1 \times 10^6$ cells/ml PBMCs were labelled with 1 μM CTV (Celltrace violet). Labelled PBMCs were pulsed with either prototype nucleocapsid whole peptide or KF9 peptide and cultured in RPMI 1640 medium (Thermo Fisher Scientific, Cat# 11875101) containing 10% FBS for 1 week. The in vitro expanded PBMCs were stained with C12/KF9 tetramers for 30 min at room temperature. After tetramer staining, surface staining with the following antibodies: CD3 BV421 (UCHT1, 1/50 dilution), CD8 APCcy7 (RPA-T8, 1/100 dilution), CD14 PerCP/Cy5.5 (HCD14, 1/100 dilution), CD19 PerCP/Cy5.5 (HIB19, 1/100 dilution), CCR7 BV510 (G043H7, 1/25 dilution), CD45RA APC (HI100, 1/25 dilution; Biolegend) was performed. Dead cells were stained with 7-aminoactinomycin D (Biolegend, Cat# 420404). After incubation for 20 min, the cells were fixed with 1% paraformaldehyde (Nacalai Tesque, Cat# 09154-85), and levels of protein expression were analysed by flow cytometry using a Cytek Northern Lights (Cytek Japan). The data obtained by flow cytometry were analysed with FlowJo v10 software (BD Biosciences).

## Retrovirus Production

Phoenix-A cells were used to produce retrovirus for transducing TCR genes into human peripheral blood mononuclear cells (PBMCs). Phoenix-A ($3.5 \times 10^5$) were cultured in a 6-well collagen-coated plate (Iwaki, Cat# 4810-010) with 2 ml of DMEM culture medium one day before transfection. TCR expression vectors were transfected into the cells using PEI Max transfection reagent (Polysciences, Cat# 24765-1) according to the manufacturer's instructions. Cells were cultured at 37 °C in a humidified atmosphere containing 5% $CO_2$. At 48 hours post-transfection, the culture supernatant was harvested and filtered through a 0.45-μm membrane filter (Merck Millipore, Burlington, MA, USA). The collected retroviral supernatant was concentrated using the Retro-X Concentrator (Takara, Cat# 631455) according to the manufacturer's instructions.

## Retroviral Transduction of TCRs into Primary CD8+ T Cells

For retroviral TCR transduction, $1 \times 10^6$ CD8+ T cells, isolated from PBMCs by using MACS with CD8 microbeads (Miltenyi Biotec, Cat# 130-045-201), were stimulated with CD3/CD28 Dynabeads™ (Thermo Fisher Scientific) in the presence of recombinant human IL-2 (PeproTech, Cat# 200-02) for two days. The PBMCs were harvested and resuspended at $5 \times 10^5$ cells/ml in RPMI 1640 containing 10% FBS in the presence of human IL-2 (30 U/ml). Meanwhile, the wells of 24-well plates were coated with 0.3 ml of RetroNectin (50 μg/ml, Takara, Kyoto, Japan) and incubated at 4 °C overnight. TCR-encoding retroviruses were adhered to the wells by centrifugation for 2 h at 2,000×g

and 32 °C. The stimulated CD8[+] T cells were then added to the virus-coated wells and centrifuged at 500×g for 10 minutes at room temperature, followed by overnight incubation at 37 °C in a 5% $CO_2$ atmosphere. The next day, the cells were transferred to freshly prepared retrovirus-coated plates and incubated at 37 °C in a 5% $CO_2$ atmosphere for another 24 hours to perform a second round of infection. Following transduction, CD8[+] T cells were expanded in the presence of IL-2 (30 U/ml) for an additional 2–3 days. RatCD2[+] CD8[+] T cells were sorted by MACS using an anti-ratCD2-FITC antibody (OX-34, BioLegend, Cat# 201303, 1:100 dilution) and anti-FITC microbeads (Miltenyi Biotec, Cat# 130-048-701). The resulting TCR-transduced CD8[+] T cells were used for functional assays on days 12–14. After tetramer staining above, surface staining with the following antibodies: ratCD2 FITC (OX-34, 1/100 dilution), CD3 BV510 (UCHT1, 1/50 dilution; Biolegend), CD8 APCcy7 (RPA-T8, 1/100 dilution) was performed. Dead cells were stained with 7-aminoactinomycin D (Biolegend, Cat# 420404). After incubation for 20 min, the cells were fixed with 1% paraformaldehyde (Nacalai Tesque, Cat# 09154-85), and levels of protein expression were analysed by flow cytometry using a Cytek Northern Lights (Cytek Japan) followed by analysis using FlowJo v10 software (BD Biosciences).

## Plasmid Construction
The plasmid expressing the SARS-CoV-2 nucleocapsid proteins of the parental (D614G-bearing B.1 lineage) was prepared using forward primer (5-AAA GGT ACC GCC GCC ACC ATG AGC GAT AA-3) and reverse primer (5-TTT GCG GCC GCT TAC TTT TCA AAC TGC GG-3). The resulting PCR fragment was digested with KpnI (New England Biolabs, Cat# R0142S) and NotI (New England Biolabs, Cat# 1089S) and inserted into the corresponding site of the pCAGGS vector. Nucleotide sequences were determined by Genetic Analyser 3500xL (Applied Biosystems), and the sequence data were analysed by GENETYX v12 (GENETYX Corporation).

## Live virus suppression assay
A live virus suppression assay by T cells was performed as previously described[20]. A549 cells expressing ACE2/C1202 (1 × 10^4 cells) were infected with each SARS-CoV-2 lineage at an MOI of 0.1 for 120 min at 37 °C. Cells were washed and cocultured with T cells at an E:T ratio of 2:1 and 1:1. Control wells containing virus-infected targets without T cells were also included. After 72 h incubation, the culture supernatant was collected and subjected to real-time RT-PCR. 5 μl of culture supernatant was lysed in an equal amount of buffer composed of 2% Triton X-100, 50 mM KCl, 100 mM Tris-HCl (pH 7.4), 40% glycerol, and 0.4 U/μl recombinant RNase inhibitor (Promega, Cat# N2615) and then incubated at room temperature for 10 min. 90 μl of RNase-free water (Nacalai Tesque, Cat# 06442-95) was added, and 3 μl of diluted sample was used as the template. Real-time RT-PCR analyses for viral RNA copy number were carried out with One Step PrimeScript™ III RT-qPCR Mix (Takara, Cat# RR600B), and reactions were performed by using LightCycler® 96 System (Roche Diagnostics GmbH, Mannheim, Germany). For the primer, Primer/Probe N2 (2019-nCoV) (Takara, Cat# XD0008) were used as follows: NIID_2019-nCOV_N_ forward, 5-AAATTTTGGGGACCAGGAAC-3; NIID_2019-nCoV_N_ reverse, 5-TGGCAGCTGTGTAGGTCAAC-3; and NIID_2019-nCoV_N_ probe, 5-FAM-ATGTCGCGCATTGGCATGGA-BHQ3. The viral RNA copy number was standardised with a Positive Control RNA Mix (2019-nCoV) (Takara, Cat#XA0142). The relative viral copy was calculated as the viral RNA copy number obtained by virus-infected targets without T cells, normalised to 1.

## pHLA protein production
The experiments were performed following our published methodology[62]. DNA plasmids in a pET30 vector encoding the HLA-C*12:02 heavy chain and β2-microglobulin were individually transformed into BL21 Escherichia coli (E. coli) competent cells (RIL strain). Each protein was expressed as inclusion bodies and purified from the transformed E. coli cells. Soluble peptide-HLA complexes were produced by refolding inclusion bodies in the following amounts: 90 mg of α-chain, 20 mg of β2-microglobulin, and 10 mg of peptide (Genscript, Piscataway, NJ, USA). The refolded mixture was dialysed into 10 mM Tris-HCl pH 8.0, and pHLA was purified using anion exchange chromatography (Cytiva, Marlborough, Massachusetts, USA).

## TCR protein production
Similarly, with HLA protein production, the experiments for TCR protein production were performed following our published methodology[6]. DNA plasmids in pET30 or pET28 vectors encoding GV37β and GV37α were individually transformed into BL21 Escherichia coli (E. coli) competent cells (RIL strain). Each protein was expressed as inclusion bodies and purified from the transformed E. coli cells. Soluble TCR complexes were produced by refolding inclusion bodies in the following amounts: 50 mg of α-chain and 50 mg of β-chain. The refolded mixture was dialysed into 10 mM Tris-HCl pH 8.0, and TCR was purified using anion exchange chromatography (Cytiva, Marlborough, Massachusetts, USA).

## Crystallisation and structure determination
Crystals of both complexes were grown via the sitting-drop, vapour diffusion method at 20 °C with a protein: reservoir drop ratio of 1:1, at a concentration of 3 mg/mL in 10 mM Tris-HCl pH 8.0, 150 mM NaCl. Crystals of HLA-C*12:02 in complex with the KF9 peptide (N_{266-274}) were grown in 2% polyethylene glycol (PEG) 400 (Sigma-Aldrich, Cat# 91893) and 20% PEG3350 (Sigma-Aldrich, Cat# 202444). Crystals of GV37 TCR-HLA-C*12:02-KF9 complex were grown in 0.2 M Magnesium formate dihydrate (Hampton Research, Cat# HR2-245) and 20% PEG3350. The crystals were cryoprotected in 30% PEG3350 added to the mother liquor, and flash-frozen in liquid nitrogen. The data were collected on the MX2 beamline at the Australian Synchrotron[63], processed using XDS[64], and scaled using the CCP4 suite[65]. Both HLA-C*12:02-KF9 structures with and without the GV37 TCR were determined by molecular replacement using the PHASER programme[66] from the CCP4 suite with a model of HLA-C*14:02 without the peptide (derived from PDB ID: 7WJ2[67]), and the GV37 TCR model was built using AlphaFold2[68]. COOT[69] was used to build in the peptide based on the electron density, and the structures were refined using PHENIX[70]. The final models have been validated and deposited using the wwPDB OneDep system, and the final refinement statistics PDB codes are summarised in Table 3. All molecular graphics representations were created using PyMOL (version 1.20; copyright, Schrodinger, LLC).

## Differential Scanning Fluorimetry
Thermal stability was measured using the ViiA 7 Real-Time PCR machine (Thermofisher, Scoresby, Australia), where pHLA samples were heated from 25 to 95 °C at a rate of 0.5 °C/min with excitation and emission channels set to yellow (excitation $549.5 \pm 10$ nm and detection at $586.5 \pm 10$ nm). The experiment was performed at two concentrations of pHLA (5 μM and 10 μM) in duplicate with 10X SYPRO Orange dye (Thermofisher, Scoresby, Australia, Cat# 6650). HLA-A*02:01-M1_{58-66} was used as a positive control[71], Fluorescence intensity data were normalised and plotted using GraphPad Prism 10 (version 10.0.3). The Tm value for each pHLA is determined to be the temperature at which 50% of maximum fluorescence intensity is reached.

## Surface plasmon resonance (SPR)
SPR experiments were conducted at 25 °C on the BIAcore T200 instrument in 10 mM Tris-HCl pH 8.0 (Thermo Fisher Scientific, Cat# BP152-5), 150 mM NaCl (Chem Supply, Cat# SA046), 0.005% surfactant P20 (Cytiva, Marlborough, Massachusetts, USA, Cat# BR100054), and

0.5% BSA (Sigma-Aldrich, St Louis, MO, USA, Cat# A7006). CM5 chips were used to immobilise GV37 TCR (coupled at ~2000 response units) in 10 mM HEPES (2-[4-(2-hydroxyethyl)piperazin-1-yl]ethanesulfonic acid) (Sigma-Aldrich, St Louis, MO, USA, Cat# H4034) and 150 mM NaCl pH 7.0. The first flow cell was loaded with HLA-A*02:01-M1 (negative control). The experiments were conducted with ten serial dilutions of the HLA-C*12:02-KF9 complex starting at 200 μM. All the experiments were conducted in duplicate (n = 2 independent experiments). BIAevaluation (v.3.1) and GraphPad Prism 10 were used for data analysis reported in Fig. 3f and 3g.

## Statistics and reproducibility

Data and statistical analysis were performed using Prism 10 (GraphPad Software). For two-way comparison, the matched-paired Wilcoxon signed-rank test (Fig. 1b, 1d, Fig. 2f, Fig. 5b, 5c, Fig. 6c, d and e) or unpaired Mann–Whitney t-test (Fig. 2e, 2f, Fig. 5e and Fig. 6b) was used.

In Fig. 2b, c, d, g, h, Fig. 3c, d, Fig. 5a–e, Supplementary Fig. 1k, Supplementary Fig. 4g, and Supplementary Fig. 5c assays were performed in triplicate. Data are representative of two or three independent experiments. In Fig. 3c, representative blots of three independent experiments were shown.

## Reporting summary

Further information on research design is available in the Nature Portfolio Reporting Summary linked to this article.

## Data availability

All data are present in this article and the Supplementary Information files. Source data are provided with this paper. All databases and datasets in this study are available from GISAID, DDBJ, IMGT, and NetMHC4.1pan (https://services.healthtech.dtu.dk/services/NetMHCpan-4.1/). The crystal structures presented here have been deposited into the PDB (https://www.rcsb.org/), and accession codes are in Table 3. Source data are provided with this paper.

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

## Acknowledgements

We would like to thank all Genotype to Phenotype Japan (G2P-Japan) Consortium members for providing plasmids, reagents, and cells, and Dr. Kazuhisa Yoshimura (Tokyo Metropolitan Institute of Public Health) and Drs. Tomohiko Takasaki and Jun-Ichi Sakuragi (Kanagawa Prefectural Institute of Public Health) for providing viruses. We thank Dr. Masafumi Takiguchi (Kumamoto University) for providing C1R-A2402, C1R-B5201, and C1R-C1202 cells. We thank Dr. David L. Woodland for assistance in editing the manuscript. We thank Kyoko Yamada for experimental support. This study was supported in part by AMED Research Programme on Emerging and Re-emerging Infectious Diseases 20fk0108539h0001 (to T.U.) and 20fk0108451s0101 (to T.U.) and AMED Research Programme on HIV/AIDS 21fk0410046 (to C.M.), and AMED Research Programme on Interdisciplinary Cutting-edge Research 23wm0325064 (to C.M.). JSPS KAKENHI Grant-in-Aid for Scientific Research B 19H03703, 22H03119 (to T.U.) and 22H02877 (to C.M., Fostering Joint International Research (A) 22KK0277(to C.M.), Scientific Research C 22K07089 (to M.T.), Takeda Science Foundation (to C.M. and M.T), and an intramural grant from Kumamoto University COVID-19 Research Projects (AMABIE) (to C.M.), IMAI MEMORIAL TRUST FOR AIDS RESEARCH (to M.T.), Shin-Nihon Foundation of Advanced Medical Research (to M.T.). DSMC is supported

by an AINSE ECR grant from ANSTO; SG was supported by an NHMRC SRF (#1159272) and is supported by an NHMRC Leadership Investigator grant (#034677). The authors would like to thank the MX team for assistance at the Australian Synchrotron. This research was undertaken in part using the MX2 beamline at the Australian Synchrotron, part of ANSTO, and made use of the ACRF detector. We thank the NIH Tetramer Core Facility (task order 57259) for providing (KF9/HLA-C*12:02) tetramers. We gratefully acknowledge all GISAID data contributors.

## Author contributions

C.M., Y.G., Y.M.A., M.T., H.H., Y.J., Y.A., T.N., Y.T, J.C.M., H.L, S.N. performed the experiments. A.Y., N.S., Y.N., R.M., T.T., N.S., H.N., Y.T. and T.S. collected clinical samples. M.K., H.K., and D.J. prepared reagents. C.M., Y.G., Y.M.A., M.T., D.S.M.C., T.U. and S.G. designed the experiments and interpreted the results. C.M., Y.M.A., J.C.M., and S.G. wrote the original manuscript. All authors reviewed and proofread the manuscript.

## Competing interests

The authors declare no competing interests.

## Additional information

[1]Division of Infection and immunity, Joint Research Center for Human Retrovirus infection, Kumamoto University, Kumamoto 8600811, Japan. [2]Department of Respiratory Medicine, Faculty of Life Sciences, Kumamoto University, Kumamoto 8608556, Japan. [3]Immunity and Infection program, La Trobe Institute for Molecular Science (LIMS), La Trobe University, Bundoora, VIC 3086, Australia. [4]Department of Biochemistry and Chemistry, School of Agriculture, Biomedicine and Environment, La Trobe University, Bundoora, VIC 3086, Australia. [5]Department of Immunology, Faculty of Medicine, Academic Assembly, University of Toyama, Toyama 9300194, Japan. [6]Department of Applied Chemistry, Faculty of Science and Engineering, Kindai University, Osaka 577-8502, Japan. [7]Center for the Study of Global Infection, Kyushu University Hospital, Kyushu University, Fukuoka 8128582, Japan. [8]Division of Infectious Diseases, Clinical Research Institute, National Hospitalization Organization, Kyushu Medical Center, Fukuoka 8108563, Japan. [9]Internal Medicine, Clinical Research Institute, National Hospital Organization, Kyushu Medical Center, Fukuoka 8108563, Japan. [10]Hematology Division, Tokyo Metropolitan Cancer and Infectious Diseases Center, Komagome Hospital, Tokyo 1138677, Japan. [11]Department of Infection Prevention and Control, Tokyo Metropolitan Cancer and Infectious Diseases Center Komagome Hospital, Tokyo 1138677, Japan. [12]Department of Infectious Disease Emergency Preparedness, Institute of Science Tokyo, Tokyo 1138510, Japan. [13]Department of Biochemistry and Molecular Biology, Monash University, Clayton, VIC 3800, Australia. [14]Department of Hematology, Rheumatology and Infectious Diseases, Kumamoto University School of Medicine, Kumamoto University Hospital, Kumamoto 8608556, Japan. [15]Department of Molecular Life Science, Tokai University School of Medicine, Isehara, Kanagawa 2591193, Japan. [16]These authors contributed equally: Yoshihiko Goto, You Min Ahn. ✉e-mail: S.Gras@latrobe.edu.au; motozono@kumamoto-u.ac.jp

