## [Transparent Peer Review file · Nature Communications]

Molecular basis of potent antiviral HLA-C-restricted CD8⁺ T cell response to an immunodominant SARS-CoV-2 nucleocapsid epitope

Corresponding Author: Professor Chihiro Motozono

Version 0:

Reviewer comments:

Reviewer #1

(Remarks to the Author)

In this manuscript, Goto et al. report the identification of an immunodominant SARS-CoV-2 nucleocapsid epitope (KF9) restricted by HLA-C in COVID-19 convalescent donors. Although the TCR repertoire elicited by KF9 is clonally diverse, KF9-specific T cells do not recognize KF9 homologous peptides from other coronaviruses, including SARS-CoV-1. However, KF9-specific T cells did suppress replication of several Omicron variants tested and maintained effector memory 12 months after infection, suggesting durable protective immunity. The immunological experiments and structure determinations are technically well done and the results justify the authors' conclusions.

Points to address:

1. TCRs GV37 and GV66 recognized the SARS-CoV-2 KF9 peptide but not the SARS-CoV-1 KF9 peptide which has Gln instead of Ala at P2. The authors show by DSF that refolded SARS-CoV-2 KF9/HLA-C complex has a T_m of 57 °C. For comparison, they should also measure the thermal stability of refolded SARS-CoV-1 KF9–HLA-C complex and determine its affinity for TCRs GV37 and GV66 by SPR.
2. The finding that CDR3 β of TCR GV37 only makes a single hydrophobic contact with the KF9 peptide is notable. Apart from the reversed TCR cited by the authors, are there TCRs with conventional docking topologies that make as few contacts with peptide as GV37? For this purpose, the authors can interrogate the TCR3d or other database of TCR–pMHC complex structures.
3. In analyzing the TCR GV37–KF9–HLA-C structure, the author should comment on why GV37 does not recognize the KF9 coronavirus sequences in Fig. 3a. Computational mutagenesis using Rosetta can be carried out to identify, at least in silico, the amino acid differences that most contribute to loss of TCR binding to pMHC or to loss of peptide binding to HLA-C.
4. What is the frequency of mutations in the KF9 epitope in the GISAID database? Are any of these mutations predicted to disrupt binding of TCR GV37 to KF9–HLA-C or of KF9 to HLA-C?

Reviewer #2

(Remarks to the Author)

The study by Goto et al. shows that the SARS-CoV-2 N protein induces efficient CTL responses in donors recovering from the early COVID-19 pandemic and identifies the N266-274 peptide as the immunodominant epitope in the most prevalent HLA haplotype in the Japanese population. This study is interesting and well conducted, but needs to broaden the scope of its findings and interest in the fields of immunology and public health.

Major issue:

The study focuses on the most common HLA haplotype in Japan. However, only 8.4% of Japanese people have this haplotype. As it has been described that many HLA-C alleles can present the same epitope, and a simple HLA binding prediction shows that KF9 can bind with high affinity to multiple HLA-C alleles, the authors must analyze whether this epitope may also be relevant in the cytotoxic immune response of other COVID-19 convalescent donors expressing these other HLA-C alleles. In addition, HLA binding prediction shows that the KF9 peptide could bind with high affinity to up to 10 other HLA-A and -B alleles, which should also be analysed.

Thus, analysis of cytotoxic response in these selected HLA-A, -B, and -C alleles would increase the relevance of the study by including a larger percentage of the Japanese population. Furthermore, as these other HLA class I alleles are relatively common in different populations around the world, this extended analysis would show the potential importance of this epitope in the global context of the cellular immune response to the coronaviral pandemic. This significantly increase the impact of this study for immunologists worldwide.

Minor points:

1. The overlapping peptide groups should be described in the Materials and Methods, as not all of them cover the full sequence of the viral proteins.
2. Lines 153-155: For example, why not pool 55-56? Why pool 25-26 but not 27-28? A rational cut-off in Figure 2a is mandatory.
3. The bioinformatics study referenced in line 167 should be explained.
4. In which peptide pool was the KF9 epitope included?
5. To confirm that the suggestion in lines 243-245 is correct, a simple HLA binding assay with the homologous peptide KQYNVTQAF should be performed.
6. Correct "Pepitde (nM)" in Figure 5a.
7. Since the KF9 epitope was conserved in the Omicron variants, the experiments in lines 318-340 are of little relevance and can be included as a supplementary figure.
8. Figure 6a: Pools 13-14, 65-66, and 95-96 showed a significant frequency of T-cell responses and responders only slightly lower than the KF9 epitope. Comment, Explanation, Relevance ...
9. In the context of Figure 6a the authors indicated in lines 349-351 that
"the KF9 epitope is still immunodominant in HLA-C*12:02+ convalescent donors even at 6 months after infection."
but since the response of KF9 is greater than that of its respective peptide pool (what?), this comparison is not correct, since the responses of pools 13-14, 65-66, and 95-96 are very similar to each other and, therefore, all are equally immunodominant.
10. Continuing with point 9, Figure 6d shows that at 12 months the response to N protein is greater than to KF9. Thus, some subdominant epitopes in the acute response generated a better T memory response than KF9. Discussion is needed.

Reviewer #3

(Remarks to the Author)

Manuscript Nr: NCOMMS-25-01918-T

Goto et al., "Molecular basis of potent antiviral 1 HLA-C-restricted CD8+ T cell response to an immunodominant SARS-CoV-2 nucleocapsid epitope"

The authors demonstrate that the nucleocapsid protein (N) of SARS-CoV-2 elicits higher CD8+ T cell responses than spike (S) and matrix (M) in Japanese convalescents. They identify an HLA-C*12:02 presented peptide epitope (KF9) that is to a large extent responsible for this dominant response. This epitope is recognized by CD8+ T cell clones with a diverse T cell receptor (TCR) repertoire that is private to the tested donors and not dominated by common public TCRs. The authors then solved the crystal structures of both the HLA class I plus peptide complex alone and bound by one of the TCRs. They observed that all solvent exposed peptide side chains were contacted by the TCR, even including P1, but without much contribution of the CDR3 of the beta TCR chain. Finally, the authors document that the HLA-C*12:02 restricted KF9 specific CD8+ T cell response is maintained up to a year after SARS-CoV-2 infection with longitudinal changes in memory subset composition. From these findings the authors conclude that KF9 specific CD8+ T cells are immunodominant in the Japanese population, maintained as memory T cells and protective in restricting SARS-CoV-2 infection by a variety of isolates.

These are interesting findings and carefully conducted studies. However, to estimate the broad protective capacity of the described T cell specificity other SARS-CoV-2 variants should probably be assessed, the sensitivity of the KF9 peptide recognition to sequence variation assessed, and the functional avidity of the respective T cells determined.

Major comments:

1. The authors tested KF9 specific CD8+ T cells against Omicron infected cells. How broadly conserved is the KF9 peptide? At least in their epitope alignment with viral gene sequences, alpha and delta SARS-CoV-2 isolates should also be queried for the conservation of this T cell epitope.
2. The authors observe an interesting pattern of TCR contacts with KF9 in their crystal structure. In order to confirm that all solvent exposed KF9 side chains contribute to TCR recognition, alanine scanning (individually replacing the TCR contacted amino acids with A) should be performed with functional T cell assays, e.g. with the TCR transduced Jurkat cells?
3. The authors speculate that the KF9 peptide variants in other coronaviruses, even SARS-CoV-1, do not stimulate the SARS-CoV-2 KF9 specific CD8+ T cells because they might not bind well to HLA-C*12:02. This should be confirmed by HLA stabilization assays on TAP deficient cell lines with HLA-C*12:02 (e.g. T2-C12).
4. In figure 3C and 5A the authors provide functional avidity assays for Jurkat cells transduced to express two KF9 specific TCRs and KF9 specific T cell lines from SARS-CoV-2 convalescents. In both instances the peptide was not titrated to high enough concentrations to reach a plateau of recognition and therefore the half-maximal functional avidity cannot be well assessed. However, the T cell lines in Figure 5 seem to have possibly 10fold higher avidity than the TCR transduced Jurkat cells and possibly a have maximal recognition avidity in the single digit nM range while the cloned TCRs seem to reach half-maximal recognition possibly only at 100nM peptide concentrations. Therefore, in order to correctly assess TCR avidity these assays should probably be repeated with titrations to higher peptide concentrations and TCR transduction into primary CD8+ T cells to correctly assess the functional avidity of the KF9 specific T cell responses and to determine if the isolated TCRs are representative in their avidity.
5. Even so the HLA-A24-B52-C12 haplotype is frequent in the Japanese population, it would be interesting if closely related HLA-C molecules, at least among the HLA-C12 alleles can present KF9 to the isolated TCRs, in order to assess if KF9 specific vaccination would also be an option in other HLA haplotypes. This could be tested with a panel of cell lines that carry closely and more distantly related HLA-C alleles.

Minor comments:

1. If the KF9 epitope is also conserved in Delta the high protection from severe disease despite near complete evasion from antibody responses in the Delta to Omicron shift could also be discussed as an indication that T cell responses provide protection against SARS-CoV-2 pathology.

Version 1:

Reviewer comments:

Reviewer #1

(Remarks to the Author)

The authors have responded satisfactorily to the previous critiques.

Reviewer #2

(Remarks to the Author)

The authors have conducted all of the experiments proposed by the reviewer and incorporated all of the suggested changes into the text. Therefore, the manuscript should be accepted.

Reviewer #3

(Remarks to the Author)

Manuscript Nr: NCOMMS-25-01918A

Goto et al., "Molecular basis of potent antiviral 1 HLA-C-restricted CD8+ T cell response to an immunodominant SARS-CoV-2 nucleocapsid epitope"

The authors demonstrate that the nucleocapsid protein (N) of SARS-CoV-2 elicits higher CD8+ T cell responses than spike (S) and matrix (M) in Japanese convalescents. They identify an HLA-C*12:02 presented peptide epitope (KF9) that is to a large extent responsible for this dominant response. This epitope is recognized by CD8+ T cell clones with a diverse T cell receptor (TCR) repertoire that is private to the tested donors and not dominated by common public TCRs. The authors then solved the crystal structures of both the HLA class I plus peptide complex alone and bound by one of the TCRs. They observed that all solvent exposed peptide side chains were contacted by the TCR, even including P1, but without much contribution of the CDR3 of the beta TCR chain. Finally, the authors document that the HLA-C*12:02 restricted KF9 specific CD8+ T cell response is maintained up to a year after SARS-CoV-2 infection with longitudinal changes in memory subset composition. From these findings the authors conclude that KF9 specific CD8+ T cells are immunodominant in the Japanese population, maintained as memory T cells and protective in restricting SARS-CoV-2 infection by a variety of isolates.

In their revised manuscript version the authors have addressed all of my concerns, namely they investigated the KF9 peptide conservation across different SARS-CoV-2 isolates, confirmed their crystal structure contacts by functional assays using alanine scanning, characterized homologous peptide binding of closely related viruses to HLA-C*1202 with TAP deficient cells, determined the EC 50 values with more complete peptide titration assays, and identified additional HLA class I molecules, mainly HLA-C*1402, that present KF9. These additions have significantly increased the quality of the manuscript.

REVIEWER COMMENTS

**Reviewer #1 (Remarks to the Author):**

In this manuscript, Goto et al. report the identification of an immunodominant SARS-
CoV-2 nucleocapsid epitope (KF9) restricted by HLA-C in COVID-19 convalescent
donors. Although the TCR repertoire elicited by KF9 is clonally diverse, KF9-specific
T cells do not recognize KF9 homologous peptides from other coronaviruses,
including SARS-CoV-1. However, KF9-specific T cells did suppress replication of
several Omicron variants tested and maintained effector memory 12 months after
infection, suggesting durable protective immunity. The immunological experiments
and structure determinations are technically well done and the results justify the
authors' conclusions.

**Our reply:**

We appreciate Reviewer 1's positive comments and are happy to hear that "*The*
*immunological experiments and structure determinations are technically well done*
*and the results justify the authors' conclusions.*"

**Points to address:**

1. TCRs GV37 and GV66 recognized the SARS-CoV-2 KF9 peptide but not the
SARS-CoV-1 KF9 peptide which has Gln instead of Ala at P2. The authors show by
DSF that refolded SARS-CoV-2 KF9/HLA-C complex has a T_m of 57 °C. For
comparison, they should also measure the thermal stability of refolded SARS-CoV-
1 KF9–HLA-C complex and determine its affinity for TCRs GV37 and GV66 by SPR.

**Our reply:**

We thank the reviewer for this suggestion. Interestingly, despite the P2-Q not being
a favored primary anchor residue for HLA-C*12:02 (P2-A/S/T/V), the binding
prediction by netMHC4.1 shows that this peptide could strongly bind to HLA-C*12:02
(% rank EL = 0.071) but 10 times lower than to SARS-CoV-2 derived KF9
(KAYNVTQAF, % rank EL = 0.006). Based on this and the reviewer's suggestion,
we successfully refolded HLA-C*12:02 with the SARS-CoV-1 derived KF9 peptide
(KF9-Q2, KQYNVTQAF). We performed a thermal stability assay with the HLA-
C*12:02-KF9-Q2 and found a 10 °C lower T_m value (47.95 ± 0.45 °C) compared to
HLA-C*12:02-SARS-CoV-2 (57.63 ± 0.48 °C) (**Supplementary Fig. 1e**), consistent
with the weak binding activity of the KF9-Q2 peptide to HLA-C*12:02 in an HLA-
stabilization assay in **the revised Supplementary Fig. 1k**. In addition, we have

performed SPR to determine the binding between GV37 TCR and HLA-C*12:02-
KF9-Q2 and found no binding, even at high TCR concentration of 400 μ M (Figure
below).

This could be due to the lower stability or a different conformation of the SARS-CoV-
1 peptide. To address this, we have predicted the structure of the SARS-CoV-1-
derived KF9 peptide (Q2) and compared it to the SARS-CoV-2-derived KF9 crystal
structure.

We have included this new information in the revised manuscript (page 9, lines 279-
382; page 40, lines 1390-1391).

2. The finding that CDR3 β of TCR GV37 only makes a single hydrophobic contact
with the KF9 peptide is notable. Apart from the reversed TCR cited by the authors,
are there TCRs with conventional docking topologies that make as few contacts with
peptide as GV37? For this purpose, the authors can interrogate the TCR3d or other
database of TCR-pMHC complex structures.

**Our reply:**

We thank the reviewer for this suggestion. We have now provided a complete
analysis of the TCRs' binding and summarised the TCR CDR3 loops' contact with
the peptide in **the revised Supplemental Table 2**. As suggested by the reviewer,
we used the list provided by the TCR3d website and analysed 42 mouse TCRpMHC
and 215 human TCRpMHC structures (a total of 257 structures). The number of
contacts was calculated using CONTACT in the CCP4 suite with 4Å as the cutoff.

The analysis showed that the lack of, or limited, contact between the CDR3 β loop
and the peptide is rare among the TCR-pMHC complex structures solved so far.
Only 10 structures show a complete absence of contact between the CDR3 β and
the peptide, and two additional structures have only one contact between the CDR3 β
and the peptide, representing 4.6% of TCRpMHC complexes. Among those 12
complexes, 3 involve a reverse mouse TCR, and interestingly, 3 are in complex with
an HLA-C allomorph. The lack of CDR3 α -peptide contact (33 complexes), or contact

limited to a single bond (4 complexes), was three-times more common than for the
CDR3 β loop. The data indicate that the CDR3 β loop is commonly involved in peptide
contact, and that the limited contact observed for GV37 TCR with the KF9 peptide is
similar to other HLA-C-specific TCRs. We have included this information in the
revised manuscript (page 10, lines 346-347; page 11, lines 348-352; page 15, lines
512-516).

3. In analyzing the TCR GV37–KF9–HLA-C structure, the author should comment
on why GV37 does not recognize the KF9 coronavirus sequences in Fig. 3a.
Computational mutagenesis using Rosetta can be carried out to identify, at least in
silico, the amino acid differences that most contribute to loss of TCR binding to
pMHC or to loss of peptide binding to HLA-C.

**Our reply:**

We show in our reply to Q1 above, that even if the HLA-C*12:02 molecule can bind
the KF9-SARS-CoV-1, the mutation at position 2 from a small Alanine to a larger Gln
was detrimental to the overall stability of the p-HLA complex. The B pocket of HLA-
C*12:02 is shallow, which won't be favorable to large side chain residues. We tested
via SPR the binding potential of the GV37 TCR to the HLA-C*12:02-KF9-Q2 complex,
and even at high concentration of the TCR (400 μ M), we could not observe TCR
binding (refer to Figure on Q1).

To understand and visualize the structure of the KF9 coronavirus-derived
homologous peptide, we have used TFold to predict the structures of those peptides
that bind to the HLA-C*12:02 allomorph.

The predicted structure of the KF9-SARS-CoV-1 peptide (purple) was generally
similar to the one observed in the crystal structure of the KF9-SARS-CoV-2 peptide
(cyan), with the single residue variation at P2, which we showed to decrease the
peptide-HLA complex stability.

The five other peptides (Figure below), derived from MERS and seasonal
coronaviruses, all have a different predicted structure in the cleft of the HLA-C*12:02
compared to KF9 derived from SARS-CoV-2.

We show that the GV37 TCR engages with the HLA-C*12:02-KF9 without disturbing
the conformation of the peptide or HLA unliganded. Therefore, it is likely that any
changes to the peptide conformation from the one observed for KF9 derived from
SARS-CoV-2, will not be, or perhaps poorly, recognized by the KF9-specific T cells.
In addition, the KF9 peptides derived from HUK1, OC43, and NL63 all have a large
side chain residue at P2, which is likely to bind poorly to HLA-C*12:02, as we show
for the KF9 derived from SARS-CoV-1 peptide. We have added an explanation of
this in the revised manuscript (page 11, lines 365-385).

**4. What is the frequency of mutations in the KF9 epitope in the GISAID database?**
**Are any of these mutations predicted to disrupt binding of TCR GV37 to KF9–HLA-**
**C or of KF9 to HLA-C?**

**Our reply:**
We thank the reviewer for this comment. We added the sequence of the KF9 epitope
region in VOCs (Alpha, Beta, Gamma and Delta) and recent variants (Omicron/XEC
and Omicron/ LP8.1) from the GISAID database in the revised manuscript (page 12,
line 420; page 41, lines 1444-1454; page 42, lines 1455-1458) and **the revised**
**Supplementary Fig. 5a**. In addition, we analysed the amino acid mutation frequency
within the KF9 epitope using 5,378,593 SARS-CoV-2 genome sequences (see
Materials and Methods for the details, page 19, lines 630-639). The frequency of
SARS-CoV-2 variants harbouring amino acid mutations within the KF9 epitope was
less than 0.05% (**revised Supplementary Fig. 5b**), indicating that this region is
highly conserved. This is consistent with previous reports that this region is crucial

for dimerization and RNA-binding in the SARS-CoV-2 nucleocapsid protein^{49, 50} and
may impact viral replication. We have added the sentence in the revised manuscript
(page 12, lines 421-423; page 13, lines 424-426; page 19 lines 630-639; page 42,
lines 1458-1462) and Dr. So Nakagawa has been added as an author to reflect the
contribution of his analysis in the revised manuscript (page 1, line 10; page 2, lines
42-43).

Of the 17 mutants presented in **Supplementary Fig. 5b**:

135 - 1 is at P1 from K to R, which should conserve the same interaction with the TCR,
so this mutant should potentially be recognised by GV37 TCR;

137 - 3 are P2 mutations from A to other small side chain residues (S/T/V) that should
be tolerated and bind to HLA-C*12:02 molecule and therefore could potentially be
recognised by the GV37 TCR;

140 - 2 at P3 from Y to F/H, all large residue with an aromatic on imidazole ring that
would help stabilise the peptide in the cleft. The P3-Y of the KF9 peptide does not
directly interact with the TCR, and F/H at P3 should enable a stable peptide. Overall,
the P3 mutations observed might not impact on GV37 TCR recognition;

144 - 2 at P4 from N to T/S, those mutations, with a frequency <0.01%, might impact and
145 decrease the GV37 TCR affinity as they both have smaller side chains compared to
146 P4-N;

147 - 2 at P5-V to L/I mutations will likely require conformational adjustment from the
148 HLA cleft, as their larger side chain would cause steric clashes. Therefore, they
potentially could have an impact on GV37 TCR recognition;

150 - 3 at P6-T to I/A/S mutations that should be tolerated by the HLA binding cleft, and
151 as the P6-T side chain does not interact with the GV37 TCR, these mutations should
not impact the GV37 TCR. The new data on Fig. 3d shows that indeed, the P6-A
mutation is well tolerated by the GV37 TCR;

154 - 3 at P7-Q to L/H/R mutations are likely to decrease the GV37 TCR recognition, as
this position is important for the TCR (Fig. 3d) and makes numerous contacts;

156 - 1 at P8-A to S should be tolerated and not impact the GV37 TCR recognition.

Overall, some of the KF9 peptide mutations are likely to decrease GV37 TCR
recognition. However, as their frequency is very low, their impact at the population
level will be limited.

**Reviewer #2 (Remarks to the Author):**

The study by Goto et al. shows that the SARS-CoV-2 N protein induces efficient CTL
responses in donors recovering from the early COVID-19 pandemic and identifies
the N266-274 peptide as the immunodominant epitope in the most prevalent HLA

haplotype in the Japanese population. This study is interesting and well conducted,
but needs to broaden the scope of its findings and interest in the fields of immunology
and public health.

**Our reply:**

We were delighted that Reviewer 2 found our study to be “*interesting and well*
*conducted*”.

**Major issue:**

The study focuses on the most common HLA haplotype in Japan. However, only
8.4% of Japanese people have this haplotype. As it has been described that many
HLA-C alleles can present the same epitope, and a simple HLA binding prediction
shows that KF9 can bind with high affinity to multiple HLA-C alleles, the authors must
analyze whether this epitope may also be relevant in the cytotoxic immune response
of other COVID-19 convalescent donors expressing these other HLA-C alleles. In
addition, HLA binding prediction shows that the KF9 peptide could bind with high
affinity to up to 10 other HLA-A and -B alleles, which should also be analysed.

Thus, analysis of cytotoxic response in these selected HLA-A, -B, and -C alleles
would increase the relevance of the study by including a larger percentage of the
Japanese population. Furthermore, as these other HLA class I alleles are relatively
common in different populations around the world, this extended analysis would
show the potential importance of this epitope in the global context of the cellular
immune response to the coronaviral pandemic. This significantly increase the impact
of this study for immunologists worldwide.

**Our reply:**

Thank you very much for this suggestion. We agree that it is important to determine
whether KF9 is presented by other HLAs, as this could be relevant for the
development of a universal vaccine. To address this, we performed an Activation-
Induced Marker assay in an additional 33 HLA-C*12:02-negative (n = 38 in total) and
10 HLA-C*12:02-positive convalescents (n = 24 in total) (62 Japanese
convalescents in total). Interestingly, we found that KF9-reactive T cells were
detected in a small number of HLA-C*12:02-negative convalescents (7 out of 38
donors) and the level of the T cell activation (median: 3.79%) in HLA-C*12:02-
negative responders is lower than that in HLA-C*12:02-positive responders (median:
33.5%) in **revised Fig. 2f**. Moreover, *in silico* analysis of HLA binding predicted that

the KF9 peptide has a strong binding activity to multiple HLA-C allomorphs within
 HLA-A, -B and -C alleles by the HLA-C*12:02-negative responders (n = 7)
 (**Supplementary Table 1**). Among them, we focused on HLA-C*14:02, most closely
 related to HLA-C*12:02, differing only by five amino acids (**Supplementary Table**
 **1**). As >20% of CD25⁺CD137⁺CD8⁺ T cells were detected in two responders (GV94
 and GV95) carrying HLA-C*14:02 (**revised Fig. 2f and revised Table 2**), we
 hypothesized that KF9 is also presented by HLA-C*14:02. We demonstrated that the
 KF9 peptide induced the production of IFN- γ by CD8⁺ T cells in a dose-dependent
 manner (**revised Fig. 2g**). The level of IFN- γ and the degranulation surface marker
 CD107a expression on CD8⁺ T cells treated with target cells expressing HLA-
 C*14:02 and N protein was higher than on mock treated cells. This difference was
 maintained at different E:T ratios (**revised Fig. 2h**). The data indicate that the KF9

Revised Fig. 2

is presented by, at least in part, HLA-C*14:02.

We have added the sentences in the revised manuscript (**page 6, lines 189-195;**
 **page 7, lines 196-227; page 26, line 926; page 37, lines 1307-1310; page 38, lines**
 **1311-1314; page 40, lines 1386-1388**) and the data in **revised Fig. 2f, 2g, 2h, and**
 **Supplementary Fig. 1d and Supplementary Table 1**). Yuka Tajima and Dr.

Hirotomo Nakata have been added as authors to reflect the contribution of their
analysis in the revised manuscript (page 1, line 6, 10; page 2, lines 39-41). Thank
you again for the important suggestion.

**Minor points:**

1. The overlapping peptide groups should be described in the Materials and Methods,
as not all of them cover the full sequence of the viral proteins.

**Our reply:**

As suggested by the reviewer, we included a description of the overlapping peptide
groups in the Materials and Methods of the revised manuscript (page 19, lines 647-
648).

2. Lines 153-155: For example, why not pool 55-56? Why pool 25-26 but not 27-28?
A rational cut-off in Figure 2a is mandatory.

**Our reply:**

We thank the reviewer for this comment. We used a cut-off value determined by
>80% as an immunodominant response. To clarify this, we have added a sentence
in the revised manuscript (page 6, line 158).

3. The bioinformatics study referenced in line 167 should be explained.

**Our reply:**

Kiyotani *et al.* conducted a bioinformatic prediction study of potential SARS-CoV-2
T cell epitopes presented by HLA molecules commonly present in the Japanese
population. We have added a sentence in the revised manuscript (page 6, lines 171-
172).

4. In which peptide pool was the KF9 epitope included?

**Our reply:**

The KF9 peptide sequence is located in the immunogenic region of the nucleocapsid
protein in convalescents carrying the *HLA-A24-B52-C12* haplotype (i.e. it is present
in both the #66 (KRTATKAYNVTQAFG) and #67 (TKAYNVTQAFGRRGP) 15-mer
peptides and in peptide pools 65-66 and 67-68. We have added a sentence to clarify
this in the revised manuscript (page 6, lines 180-184).

5. To confirm that the suggestion in lines 243-245 is correct, a simple HLA binding
assay with the homologous peptide KQYNVTQAF should be performed.

**Our reply:**

We thank the reviewer for this suggestion. To determine whether the homologous
SARS-CoV-1-derived KQYNVTQAF (KF9-Q2) binds to HLA-C*12:02, we generated
TAP1/2-deficient cell lines expressing HLA-C*12:02 (MCF7ΔTAP1/2-C1202). An
HLA stabilization assay revealed that the KF9-Q2 peptide weakly bound to HLA-
C*12:02. We have added the sentence in the revised manuscript (page 9, lines 302-
305; page 22, lines 743-762; page 26, line 928; page 40, lines 1397-1403) and the
data in **the revised Supplementary Figure 1i-k.**

6. Correct "Pepitde (nM)" in Figure 5a.

**Our reply:**

We sincerely apologize for the mistake. We have corrected it in **the revised Figure**
**5a.**

7. Since the KF9 epitope was conserved in the Omicron variants, the experiments in
lines 318-340 are of little relevance and can be included as a supplementary figure.

**Our reply:**

As suggested by the reviewer, we replaced the data on the suppression of viral
replication of Omicron BA.1, BA.2, and BA. 5 with **the revised Supplementary Fig.**
**5d.**

8. Figure 6a: Pools 13-14, 65-66, and 95-96 showed a significant frequency of T-cell
responses and responders only slightly lower than the KF9 epitope. Comment,
Explanation, Relevance ...

**Our reply:**

We thank the reviewer for this comment. This is due to the differences in the length
and concentration of the peptides. As we mentioned earlier, the KF9 peptide
sequence is included in 15-mer peptides (#66: KRTATKAYNVTQAFG and #67:
TKAYNVTQAFGRRGP), they are not optimal epitopes like the 9-mer KAYNVTQAF
peptide. It is unlikely that long peptides can be accommodated on the peptide binding
groove of HLA class I molecules with their central region bulged or either end

extended away. Alternatively, long peptides would be digested exogenously or
endogenously and loaded onto the empty HLA-C*12:02 in the endoplasmic reticulum,
and the resultant KF9 peptide/HLA-C*12:02 complexes could be presented on the
cell surface. Regarding this point, the concentration of the KF9 peptide on the cell
surface would be lower than that of the KF9 peptide. We have reported this
previously (Motozono et al. *J. Immunol.*, 2009, PMID: 19380801).

To clarify this, we have added the concentration of the peptides in the legend of **Fig.**
**6a** in the revised manuscript (page 39, lines 1371-1372).

9. In the context of Figure 6a the authors indicated in lines 349-351 that

“the KF9 epitope is still immunodominant in HLA-C*12:02+ convalescent donors
even at 6 months after infection.”

but since the response of KF9 is greater than that of its respective peptide pool
(what?), this comparison is not correct, since the responses of pools 13-14, 65-66,
and 95-96 are very similar to each other and, therefore, all are equally
immunodominant.

**Our reply:**

We agree with this point. We should compare the response to overlapping peptides
equally as the concentration of the overlapping peptides was lower than that of the
KF9, as described in the legend of Fig. 6a. We have revised the sentence
accordingly in the revised manuscript (page 13, lines 448-449).

10. Continuing with point 9, Figure 6d shows that at 12 months the response to N
protein is greater than to KF9. Thus, some subdominant epitopes in the acute
response generated a better T memory response than KF9. Discussion is needed.

**Our reply:**

We apologize for our misleading description in **Fig. 6d**. We detected proliferated
CTV^{low} KF9-specific T cells by KF9/C12 tetramers upon stimulation of either
nucleocapsid overlapping peptide pools and the KF9 peptide after 6- and 12-months
post-infection to investigate the contribution of KF9-specific T cells among SARS-
CoV-2 N-reactive T cells. To clarify this, we have added the sentence in the revised
manuscript (page 14, lines 467 and 470-473).

**Reviewer #3 (Remarks to the Author):**

The authors demonstrate that the nucleocapsid protein (N) of SARS-CoV-2 elicits
higher CD8+ T cell responses than spike (S) and matrix (M) in Japanese
convalescents. They identify an HLA-C*12:02 presented peptide epitope (KF9) that
is to a large extent responsible for this dominant response. This epitope is
recognized by CD8+ T cell clones with a diverse T cell receptor (TCR) repertoire that
is private to the tested donors and not dominated by common public TCRs. The
authors then solved the crystal structures of both the HLA class I plus peptide
complex alone and bound by one of the TCRs. They observed that all solvent
exposed peptide side chains were contacted by the TCR, even including P1, but
without much contribution of the CDR3 of the beta TCR chain. Finally, the authors
document that the HLA-C*12:02 restricted KF9 specific CD8+ T cell response is
maintained up to a year after SARS-CoV-2 infection with longitudinal changes in
memory subset composition. From these findings the authors conclude that KF9
specific CD8+ T cells are immunodominant in the Japanese population, maintained
as memory T cells and protective in restricting SARS-CoV-2 infection by a variety of
isolates.

These are interesting findings and carefully conducted studies. However, to estimate
the broad protective capacity of the described T cell specificity other SARS-CoV-2
variants should probably be assessed, the sensitivity of the KF9 peptide recognition
to sequence variation assessed, and the functional avidity of the respective T cells
determined.

**Our reply:**

We were delighted that this reviewer feels that “*These are interesting findings and*
*carefully conducted studies*”.

**Major comments:**

1. The authors tested KF9 specific CD8+ T cells against Omicron infected cells. How
broadly conserved is the KF9 peptide? At least in their epitope alignment with viral
gene sequences, alpha and delta SARS-CoV-2 isolates should also be queried for
the conservation of this T cell epitope.

**Our reply:**

We thank the reviewer for this comment. We added the sequence of the KF9 epitope
region in VOCs (Alpha, Beta, Gamma and Delta) and recent variants (Omicron/XEC
and Omicron/ LP8.1) from the GISAID database in the revised manuscript (**page 12,**

line 420; page 41, lines 1444-1454; page 42, lines 1455-1458) and **the revised**
**Supplementary Fig. 5a**. In addition, we analysed the amino acid mutation frequency
within the KF9 epitope using 5,378,593 SARS-CoV-2 genome sequences (see
Materials and Methods for the details, **page 19, lines 630-639**). The frequency of
SARS-CoV-2 variants harbouring amino acid mutations within the KF9 epitope was
less than 0.05% (**revised Supplementary Fig. 5b**), indicating that this region is
highly conserved. This is consistent with previous reports that this region is crucial
for dimerization and RNA-binding in the SARS-CoV-2 nucleocapsid protein^{49, 50} and
may impact viral replication. We have added the sentence in the revised manuscript
(**page 12, lines 421-423; page 13, lines 424-426; page 19 lines 630-639; page 42,**
**lines 1458-1462**) and Dr. So Nakagawa has been added as an author to reflect the
contribution of his analysis in the revised manuscript (**page 1, line 10; page 2, lines**
**42-43**).

2. The authors observe an interesting pattern of TCR contacts with KF9 in their
crystal structure. In order to confirm that all solvent exposed KF9 side chains
contribute to TCR recognition, alanine scanning (individually replacing the TCR
contacted amino acids with A) should be performed with functional T cell assays, e.g.
with the TCR transduced Jurkat cells?

**Our reply:**

According to the reviewer's suggestion, alanine scanning except position 2 and 8
using Jurkat cells expressing GV37 and GV66 TCR (**the revised Fig. 3d**). We
confirmed that GV37 TCR were not tolerant toward alanine-substitutions of the
exposed amino acids at position 4, 5 and 7, consistent with their crystal structure
(**Fig. 4d**). We have added the sentence to this effect in the revised manuscript (**page**
**8, lines 268-271; page 9; lines 272-275 and 297-298; page 26, line 928; page 38;**
**lines 1328-1330**).

3. The authors speculate that the KF9 peptide variants in other coronaviruses, even
SARS-CoV-1, do not stimulate the SARS-CoV-2 KF9 specific CD8+ T cells because
they might not bind well to HLA-C*12:02. This should be confirmed by HLA
stabilization assays on TAP deficient cell lines with HLA-C*12:02 (e.g. T2-C12).

**Our reply:**

To investigate whether the homologous SARS-CoV-1-derived KQYNVTQAF (KF9-
Q2) binds to HLA-C*12:02, we generated TAP1/2-deficient cell lines expressing
HLA-C*12:02 (MCF7 Δ TAP1/2-C1202). An HLA stabilization assay revealed that the

KF9-Q2 peptide weakly bound to HLA-C*12:02. We have added the sentence in the
revised manuscript (page 9, lines 302-305; page 22, lines 743-762; page 26, line
928; page 40, lines 1397-1403) and the data in **the revised Supplementary Figure**
**1i-k.**

4. In figure 3C and 5A the authors provide functional avidity assays for Jurkat cells
transduced to express two KF9 specific TCRs and KF9 specific T cell lines from
SARS-CoV-2 convalescents. In both instances the peptide was not titrated to high
enough concentrations to reach a plateau of recognition and therefore the half-
maximal functional avidity cannot be well assessed. However, the T cell lines in
Figure 5 seem to have possibly 10fold higher avidity than the TCR transduced Jurkat
cells and possibly a have maximal recognition avidity in the single digit nM range
while the cloned TCRs seem to reach half-maximal recognition possibly only at
100nM peptide concentrations. Therefore, in order to correctly assess TCR avidity
these assays should probably be repeated with titrations to higher peptide
concentrations and TCR transduction into primary CD8⁺ T cells to correctly assess
the functional avidity of the KF9 specific T cell responses and to determine if the
isolated TCRs are representative in their avidity.

**Our reply:**

According to the reviewer's suggestion, we performed further peptide titrations of
KF9-specific T-cells from 0.01 nM to 100 nM using T cell lines from other donors
(GV88, GV90, and GV96). The EC₅₀ values of the peptides were calculated as the
concentration of peptide that exhibited a half-maximal activation of T cell lines with
stimulation of 100 nM (a plateau of recognition) defined as maximal. The EC₅₀ values
of T cell lines from GV88, GV90, and GV96 were 3.39 ± 0.26, 5.58 ± 0.23, and 6.53
± 0.97, respectively. To evaluate the TCR avidity of the GV37 TCR, we generated
GV37 TCR-transduced CD8⁺ T cells by retroviral transduction and performed peptide
titrations from 0.01 nM to 1000 nM. The EC₅₀ value of the peptide was calculated as
the concentration of peptide that exhibited a half-maximal activation of T cell lines
with stimulation of 1000 nM (a plateau of recognition) defined as maximal and was
4.41 ± 0.3 (**revised Supplementary Fig. 4g**). These data demonstrated that the
sensitivity of GV37 TCR-transduced CD8⁺ T cells is comparable to that of parental
T cell lines. We also repeated peptide titrations of Jurkat cells expressing GV37 TCR
and GV66 TCR from 0.3 nM to 1000 nM, but could not calculate the EC₅₀ value in
this assay (**revised Fig. 3c**).

We have added sentences in the revised manuscript to describe these finding (page
12, line 390 and lines 395-401; page 23, lines 783-794 and lines 796-815; page 24,
line 816-822; page 26, line 929; page 41, line 1430, lines 1434-1438 and line 1440)
and replaced the representative data in **the revised Fig. 3c, Fig. 5a and 5b** and
**Supplementary Fig. 4b and 4d**, and added the data of GV37 TCR-transduced CD8⁺
T cells in **Supplementary Fig. 4 e-g**. Yan Jin has been added as an author to reflect
the contribution of their analysis in the revised manuscript (page 1, line 5).

5. Even so the HLA-A24-B52-C12 haplotype is frequent in the Japanese population,
it would be interesting if closely related HLA-C molecules, at least among the HLA-
C12 alleles can present KF9 to the isolated TCRs, in order to assess if KF9 specific
vaccination would also be an option in other HLA haplotypes. This could be tested
with a panel of cell lines that carry closely and more distantly related HLA-C alleles.

**Our reply:**

Thank you very much for this suggestion. We agree that it is important to determine
whether KF9 is presented by other HLAs, as this could be relevant for the
development of a universal vaccine. To address this, we performed an Activation-
Induced Marker assay in an additional 33 HLA-C*12:02-negative (n = 38 in total) and
10 HLA-C*12:02-positive convalescents (n = 24 in total) (62 Japanese
convalescents in total). Interestingly, we found that KF9-reactive T cells were
detected in a small number of HLA-C*12:02-negative convalescents (7 out of 38
donors) and the level of the T cell activation (median: 3.79%) in HLA-C*12:02-
negative responders is lower than that in HLA-C*12:02-positive responders (median:
33.5%) in **revised Fig. 2f**. Moreover, *in silico* analysis of HLA binding predicted that
the KF9 peptide has a strong binding activity to multiple HLA-C allomorphs within
HLA-A, -B and -C alleles by the HLA-C*12:02-negative responders (n = 7)
(**Supplementary Table 1**). Among them, we focused on HLA-C*14:02, most closely
related to HLA-C*12:02, differing only by five amino acids (**Supplementary Table**
**1**). As >20% of CD25⁺CD137⁺CD8⁺ T cells were detected in two responders (GV94
and GV95) carrying HLA-C*14:02 (**revised Fig. 2f and revised Table 2**), we
hypothesized that KF9 is also presented by HLA-C*14:02. We demonstrated that the
KF9 peptide induced the production of IFN- γ by CD8⁺ T cells in a dose-dependent
manner (**revised Fig. 2g**). The level of IFN- γ and the degranulation surface marker
CD107a expression on CD8⁺ T cells treated with target cells expressing HLA-
C*14:02 and N protein was higher than on mock treated cells. This difference was

maintained at different E:T ratios (**revised Fig. 2h**). The data indicate that the KF9
 is presented by, at least in part, HLA-C*14:02.

Revised Fig. 2

We have added the sentences in the revised manuscript (**page 6, lines 189-195;**
 **page 7, lines 196-227; page 26, line 926; page 37, lines 1307-1310; page 38, lines**
 **1311-1314; page 40, lines 1386-1388)** and the data in **revised Fig. 2f, 2g, 2h, and**
 **Supplementary Fig. 1d and Supplementary Table 1**). Yuka Tajima and Dr.
 Hirotomo Nakata have been added as authors to reflect the contribution of their
 analysis in the revised manuscript (**page 1, line 6, 10; page 2, lines 39-41**). Thank
 you again for the important suggestion.

**Minor comments:**

1. If the KF9 epitope is also conserved in Delta the high protection from severe
 disease despite near complete evasion from antibody responses in the Delta to
 Omicron shift could also be discussed as an indication that T cell responses provide
 protection against SARS-CoV-2 pathology.

**Our reply:**

We thank the reviewer for this interesting suggestion. The KF9 peptide sequence is
conserved not only in the Delta variant but also in other variants. It would be
interesting to determine whether a potent KF9-specific T cell response against
variants would be observed in convalescents infected with SARS-CoV-2 variants,
which is associated with clinical outcome (severity, symptomatic or asymptomatic).
We have started collecting PBMC samples from convalescents with clinical data
(HLA alleles, severity, etc) in large Japanese cohorts to address this question in
future studies. Accordingly, we have added a sentence in the revised manuscript
(page 16, lines 544-548).